



**1 Skin Sea Surface Temperature schemes in coupled ocean-**
**2 atmosphere modeling: the impact of chlorophyll-interactive e-folding**
**3 depth.**

Vincenzo de Toma[2], Daniele Ciani[1], Yassmin Hesham Essa[1,3,4], Chunxue Yang[1], Vincenzo Artale[1],
Andrea Pisano[1], Davide Cavaliere[1], Rosalia Santoleri[1], and Andrea Storto[1]
[1] CNR-ISMAR, Consiglio Nazionale delle Ricerche, Istituto di Scienze Marine, via
Fosso del Cavaliere 100, 00133 Rome, Italy.
[2] CNR-ISMAR, Consiglio Nazionale delle Ricerche, Istituto di Scienze Marine, Calata Porta Di Massa - Porto Di Napoli
80, 80133 Naples, Italy.
[3] GUF-IAU, Goethe University Frankfurt, Institut fuer Atmosphaere und Umwelt, Frankfurt, Germany
[4] ARC-CLAC, Agricultural Research Center, Central Laboratory for Agricultural Climate, Giza, Egypt
*Correspondence to*: Vincenzo de Toma (vincenzo.detoma@cnr.it)
**Abstract.** In this paper, we explore different prognostic methods to account for skin sea surface temperature
diurnal variations in a coupled ocean-atmosphere regional model of the Mediterranean Sea. Our aim is to
characterize the sensitivity of the considered methods with respect to the underlying assumption of how the
solar radiation shapes the warm layer of the ocean. All existing methods truncate solar transmission coefficient
at a constant warm layer reference depth; instead, we develop a new scheme where this latter is estimated from
a chlorophyll dataset as the e-folding depth of solar transmission. This allows spatial and temporal variations
of the warm layer extent to depend on seawater transparency. Comparison against satellite data shows that our
new scheme improves the diurnal signal especially during winter, spring, and autumn, with an averaged bias
on monthly scales year-round smaller than 0.1 *K*. In April, when most of the drifters' measurements are
available, the new scheme mitigates the bias during nighttime, keeping it positive but smaller than 0.12 *K* during
the rest of the monthly-averaged day. The new scheme implemented within the ocean model improves the old
one by about 0.1 *K*, particularly during June. All the methods considered here showed differences with respect
to objectively analyzed profiles confined between 0.5 K during winter and 1 *K* in summer for both the eastern
and the western Mediterranean regions, especially over the uppermost 60 *m*. Overall, the surface net total heat
flux shows that the use of a skin SST parametrization brings the budget about 1.5 *W/m²* closer to zero on an
annual basis, despite all simulations showing an annual net heat loss from the ocean to the atmosphere. Our
"chlorophyll-interactive" method proved to be an effective enhancement of existing methods, its strength
relying on an improved physical consistency with the solar extinction implemented in the ocean component.

## 1 Introduction

Air-sea fluxes govern the energy exchange at the ocean-atmosphere interface. A reliable representation of
the Sea Surface Temperature (SST) diurnal cycle, i.e. the typical SST oscillation/excursion between night and
day mainly due to solar heating, is crucial to accurately estimate air-sea heat fluxes (Kawai and Wada, 2007,
Soloviev and Lukas, 2013), whose direct measurement is very difficult. Indeed, diurnal warming events can
often exceed 5 K depending on weather conditions (Soloviev and Lukas, 1997) and geographical location,



typically at tropical and mid-latitudes but also occasionally at high latitudes (Karagali and Høyer, 2013). Large
diurnal warming events can lead to changes in air-sea heat flux locally reaching up to $60 W/m^2$ (Fairall et al.,
1996, Ward, 2006, Kawai and Wada, 2007, Marullo et al., 2010, Marullo et al., 2016) on a variety of scales,
ranging from the short regional ocean weather ones to large seasonal or long-term ones.

Therefore, there is a wide interest in the development of models to accurately reconstruct SST diurnal variations
in order to improve the representation of air-sea energy exchanges, especially, but not solely, within the coupled
ocean-atmosphere modeling framework (Penny et al., 2019).
The net energy flux across the air-sea interface results from four contributions: the net solar radiation; latent
and sensible heat fluxes, and the net thermal radiation. The last three contributions depend on SST and have a
direct impact in determining ocean heat uptake or dynamical processes such as deep-water formation (Chen
and Houze Jr., 1997). Ideally, the most accurate flux estimate would imply the knowledge of the temperature
right at the atmosphere-ocean separation interface. From an observational point of view, the skin SST is the
temperature immediately adjacent to the ocean surface (~10-20 microns depth) that is measurable, typically
from infrared radiometers, and thus a key parameter to understand heat flux exchange (Minnet et al., 2019).
Indeed, following what is measurable by current sensors, the GHRSST-PP (i.e. the Global ocean data
assimilation experiment High Resolution SST Pilot Project) introduced the distinction between skin, sub-skin,
depth, and foundation SST (Donlon et al., 2007), which can be respectively regarded as successive, better-to-
worse approximations to the ideal target, i.e. SST right at the interface, which is actually impossible to measure.
However, in most of the widely used ocean models and configurations, the too-coarse vertical resolution does
not allow to direct modeling skin SST (the first model layer being only around 0.5 - 1 meter thick, e. g. the
ocean model NEMO). Therefore, one must use schemes to reconstruct skin SST variations. Sadly, the only
thing one can be sure about is that in general no model will be able to perfectly reproduce skin SST diurnal
variations, and there are different ways to approach this challenging problem, each one still with its own
limitations (see Kawai and Wada, 2007 and references therein). Simplified models widely employed in ocean
and atmosphere state-of-the-art models parameterize the skin SST dynamics via the distinction of two main
effects: the cool skin and the warm layer. Due to its interactions with the atmosphere, the temperature right at
the ocean surface is supposed to be almost anywhere and anytime cooler than the ones below, resulting in the
ocean being covered with a cool skin layer: one of the very first and simpler models assumes this cool skin
temperature difference as proportional to the ratio between heat fluxes and kinematic stress (Saunders, 1967),
via the Saunders' constant.
The cool skin effect is very important in obtaining accurate estimates of the latent and sensible heat flux,
especially because its consideration modifies specific humidity at the ocean surface, which is one of the factors
in the bulk formula. Indeed, latent and sensible heat fluxes are defined as the heat transfer across the
ocean/atmosphere interface due to turbulent air motions (the former including the one resulting from
condensation or evaporation). For example, a recent study in the South China Sea showed that during nighttime
the cool skin temperature difference is around 1 $K$, and there's currently a large uncertainty in the Saunders'





constant (Zhang et al., 2020). A warm layer (in which diurnal warming effectively takes place) develops below
this cool skin, and its extent reaches a depth at which the penetration of solar radiation can be neglected (usually
fixed to 3m by most of existing parameterizations – see section 3.3 for more details). Diurnal warm layer
anomalies (which can sometimes exceed 3K) can potentially impact both the atmosphere and ocean mean state
on a variety of spatial (ranging from regional, basin-wide to global ones) and temporal scales (relevant for
weather or seasonal forecast to long-term climatic trends) (Donlon et al., 2007). The skin SST diurnal warming
amplitude increases under low surface winds (smaller than 2 $m/s$) and intense solar radiation (higher than
typical daily peaks, around 900 $W/m^2$) conditions, smaller in winter and at the poles than in summer and in
the tropics. The accuracy of skin SST models, and therefore their ability to reconstruct skin SST diurnal
variations is crucial especially in heat budget closure problems, which are still a subject of active debate
especially in climate change hot spot regions such as the Mediterranean domain (see Marullo et al., 2021 and
references therein). Skin SST schemes are also crucial for assimilating daytime SST data from satellite sensors
(Penny et al., 2019; Storto and Oddo, 2019, Jansen et al., 2019), with obvious impact on the accuracy of
numerical weather and ocean predictions; a correct account of skin SST diurnal variations in turn is crucial for
flux calculations, which is already a very delicate problem also from an instrumental point of view.
Our main aim here is therefore to improve existing skin SST prognostic schemes, investigating the impact of
accounting for seawater's transparency conditions in modeling solar radiation extinction in the upper ocean.
The paper is structured as follows: after this introduction, we describe the data and coupled modeling system
in section 2. The mathematical context in which we developed our new method, whose novelty stands in
allowing the warm layer's extent to vary in space and time according to a chlorophyll-concentration climatology
follows in section 3. In section 4 we present results, discussing them and drawing conclusions in section 5.

## 98  2    Data and Modeling System

We describe here the data and the coupled regional modeling system used in this study. Our description here
is functional to the scope of this paper, and far from a complete depiction of each dataset. We redirect the
documentation and the appropriate literature describing each data and model in depth.

### 102  2.1  Operational MED DOISST within CMEMS

The MEDiterranean Diurnal Optimally Interpolated Sea Surface Temperature (MED DOISST) product,
operationally distributed and freely available within the Copernicus Marine Environmental Service (CMEMS)
provides gap-free (L4) hourly mean maps of sub-skin SST at $1/16^o$ horizontal resolution over the Mediterranean
domain, covering from 2019 to present. Sub-skin SST is defined as the temperature at the base of the cool skin
layer, typically sensed by microwave radiometers, and representative of a depth of few millimeters from the
ocean's surface (Minnet et al., 2019).

This product combines satellite data acquired from the Spinning Enhanced Visible and InfraRed Imager
(SEVIRI) and model data from the Mediterranean Forecasting System (MedFS), respectively used as




observations and first guess for an optimal interpolation, giving a L4 field representative of subskin SST (see
Pisano et al., 2022 and references therein). In all diagnostics involving these data (and presented in the following
sections), regions where the percentage of model data is higher than 50% have been masked out both in
CMEMS MED DOISST and our experiments.
**2.2    iQuam in-situ data**
SST from drifter data were used for validation purposes and acquired from the iQuam (In situ SST Quality
Monitor) archive (Xu and Ignatov, 2014). The iQuam provides high-quality and quality controlled (QC) in-situ
SST data collected from various platforms, such as drifters, Argo Floats, ships, tropical and coastal moored
buoys. iQuam SST data are also provided along with quality level flags ranging from 0 to 5, with 5
corresponding to the highest quality level (Xu and Ignatov, 2014). For this study, SST with quality level equal
five were selected from drifters only, since they provide the temperature measurement closest to the surface
(compared to the other available instruments), ranging between 20-30$cm$ (depending on the drifter type).
Additionally, we interpolated model outputs on drifters' location in time and space. Table S1 resumes the
number of available measurements for each given month and hour of the day. A total number of 555919 records
were available after the quality flag and platform selection, with the month of April being the most populated
one, with 222996 measurements, and 10361 measurements at 9:00 am.

**2.3    EN4 objective analysis**
EN4, the quality controlled subsurface ocean temperature and salinity profiles and objective analyses, were
used to assess the impact on the temperature vertical profiles. To facilitate the comparison, we made use of the
objective analyses after bias corrections of Expendable Bathythermograph (XBT) calibrations (Gouretski and
Reseghetti, 2010, Gouretski and Cheng, 2020), which give a gridded version of the dataset on a 1-degree regular
grid. In the comparison, model outputs were interpolated on this grid.
**2.4    Mediterranean Chlorophyll concentration**
Chlorophyll data were used to estimate e-folding depths' seasonality (see Methods, Section 3). These data
are a daily interpolation at 0.3 km horizontal resolution over the Mediterranean domain, and result from a
merging between multiple sensors (MERIS - MEdium Resolution Imaging Spectrometer from ESA, SeaWiFS
- Sea-viewing Wide Field-of-view Sensor and MODIS - Moderate Resolution Imaging Spectroradiometer from
NASA, VIIRS - Visible Infrared Imager Radiometer Suite from NOAA, and most recently the Copernicus
Sentinel 3A OLCI - Ocean and Land Colour Instrument), as detailed in the product description (see Volpe et
al., 2019 and references therein for further details).



## 2.5 ECMWF Atmospheric Reanalysis - ERA5
We used heat fluxes (net solar radiation, latent and sensible heat fluxes, net thermal radiation) from ERA5
at 0.25° horizontal and hourly temporal resolution (Hersbach et al., 2020) as reference for comparing
performances across simulations with different skin SST schemes. Despite their possible biases in air-sea fluxes,
atmospheric reanalyses at day are still widely thought to provide the best gap-free and dynamically consistent
reconstructions of the atmosphere system (Valdivieso et al., 2017, Storto et al., 2019).
## 2.6 Mixed Layer Depth 1969-2013 Climatology

Data from a mixed layer depth (MLD) climatology was used to test to what extent our modified scheme
correctly represents the seasonality of the mixed layer.
This monthly gridded climatology was produced using MBT, XBT, Profiling floats, Gliders, and ship-based
CTD (Conductivity, Temperature, Depth) data from different databases and carried out in the Mediterranean
Sea between 1969 and 2013. As for the model outputs, MLD is calculated with a $\Delta T = 0.1°C$ criterion relative
to 10m reference level on individual profiles (Houpert et al., 2015a, Houpert et al., 2015b).

## 2.7 ISMAR Mediterranean Earth System Model (MESMAR)
MESMAR is a newly developed coupled regional modeling framework for the Mediterranean region (Storto
et al., 2023). MESMAR includes the following components:
• the ocean model: NEMO v4.0.7, with horizontal resolution of about 7 $km$, 72 vertical levels and a timestep
of 7.5 minutes (NEMO System Team, 2019);
• the atmosphere model: WRF v4.3.3, with 41 vertical hybrid levels and horizontal resolution of about 15
$km$, covering the European branch of the international Coordinated Downscaling Experiment (EURO-
CORDEX) domain, and a timestep of 1 minute (Skamarock et al., 2019);
• an interactive runoff model: HD v5.0.1, with a timestep of 30 minutes and 1/12° degree horizontal
resolution over Europe (Hagemann et al., 2020);
• the coupler: OASIS3-MCT, coupling the three models with a coupling frequency of 30 minutes, and
using the SCRIP library to interpolate fields between different model grids (Craig et al., 2017) ;
We report in figure 1 a graphical summary of different grids. Further details of its implementation, tuning, and
performances are described in (Storto et al., 2023).



## 3 Methods

Many schemes to reconstruct the skin SST diurnal variations rely on the existence of a cool skin and a warm
layer, respectively in the upper micrometers and few meters of the ocean, whose dynamics strongly depends on
wind conditions and solar radiation extinction within the upper ocean. To explain the rationale behind the
developments in our new method, we need to recap here some elements of this theory, which is mostly based
on Zeng and Beljaars, 2005 (named ZB05 hereafter) work.
We start from the one-dimensional heat transfer equation in the ocean:

$$\frac{\partial T}{\partial t} = \frac{\partial}{\partial z}\left(K_w + k_w\right)\frac{\partial T}{\partial z} + \frac{1}{\rho_w c_w}\frac{\partial R}{\partial z} \tag{1}$$

in which the subscript $w$ refers to water properties, $T$ is seawater temperature, $K_w$ is the turbulent diffusion
coefficient, $k_w$ is the molecular thermal conductivity, $\rho_w$, $c_w$ are respectively seawater density and heat capacity
per unit volume, $R$ is the net solar radiation flux, defined as positive downward.

### 3.1 Cool Skin

We assume that there exists an oceanic molecular sublayer of depth $\delta$, where $K_w$ is negligible, and
temperature can be assumed constant in time, since it is always cooler than temperature of the underlying
seawater (Donlon et al., 2007, Zeng and Beljaars, 2005). Then integration of eqn. (1) gives, $\forall z \in [0, -\delta]$

$$k_w \frac{\partial T}{\partial z} + \frac{1}{\rho_w c_w}[R(z) - R_s] - \underbrace{k_w \frac{\partial^2 T}{\partial z^2}}_{\mathcal{O}(z^2)} = const, \tag{2}$$

where $R_s$ is solar radiation at the surface, assuming this constant to be the top boundary condition at $z = 0$:

$$\rho_w c_w k_w \frac{\partial T}{\partial z}\bigg|_{z=0} = Q = LH + SH + LW, \tag{3}$$

in which *LH, SH, LW* are respectively the surface fluxes of latent, sensible heat and net
long wave radiation.
Thus, eqn. (2) can be rewritten as

$$\rho_w c_w k_w \frac{\partial T}{\partial z} = Q + R_s - R(z). \tag{4}$$

Making a further integration we get the cool skin temperature difference:

$$T_s - T_{-\delta} = \frac{\delta}{\rho_w c_w k_w}\left(Q + f_s R_s\right), \tag{5}$$

where $T_s$ and $T_{-\delta}$ are respectively the temperature at the upper (air-sea interface) and lower limits of the cool
skin layer, while $f_s$ is the fraction of solar radiation absorbed in this layer:





$$f_s = \frac{1}{\delta} \int_{-\delta}^{0} \left( 1 - \frac{R(z)}{R_s} \right) \, dz,$$

which depends on the way radiation gets absorbed within the cool skin.

Eq. (5) is analogous to Saunders' model. Indeed, Saunders, 1967 was one of the first to construct a theory for the ocean "cool skin" effect (already known from decades at those times), i.e. the observed temperature at the air-sea interface is generally cooler than the temperature of the water at about 10 cm depth, especially during nighttime. This effect takes place mainly because of the transfer of energy between the ocean and the atmosphere, realized via heat loss and momentum transfers (wind stress). In a nutshell, at the end of its derivation (Saunders, 1967), he obtains the following expression for the temperature difference across the cool skin, $\Delta T_c$:

$$\Delta T_c \; = \; \lambda \frac{Q \nu_w}{k_w (\tau/\rho_w)^{1/2}}, \qquad (6)$$

where $\lambda$ is the Saunders' proportionality constant, $Q$ has already been defined above, $\tau/\rho_w$ is the kinematic stress (ratio between wind stress module and seawater density), and $\nu_w$, $k_w$ are respectively the kinematic viscosity and thermal conductivity of seawater. Saunders' formulation was originally conceived for low, nonzero wind conditions and neglecting the effect of solar radiation. As noticed by Artale et al., 2002 (named A02 hereafter), with a constant $\lambda$, eqn. (6) becomes problematic in limiting cases of low and very high wind speeds (greater than 7 $m/s$). Thus, they proposed to include a wind dependence in Saunders' constant, in order to still have a finite, nonzero cool skin to bulk temperature difference even when the wind speed goes to zero or becomes very high.

This scheme has proven to have good performances compared to other schemes also on a mooring site in the Pacific Ocean (Tu and Tsuang, 2005).

## 3.2   Warm Layer

Below the skin layer, turbulent transfer is much more effective, and $k_w$ can be neglected in favor of $K_w$. Integrating eqn. (1) within the $[-d, -\delta]$ layer, we get:

$$\frac{\partial}{\partial t} \int_{-d}^{-\delta} T \, dz = \frac{Q + R_s - R(-d)}{\rho_w c_w} - K_w \frac{\partial T}{\partial z}\bigg|_{z=-d}, \qquad (7)$$

where $d$ is a reference depth which can be assumed as the depth at which the diurnal cycle can be omitted.

The turbulent diffusion coefficient can be expressed as (Large et al., 1994):

$$K_w = k u_{*w} (-z) / \phi_t \left( \frac{-z}{L} \right), \qquad (8)$$



in which $k = 0.4$ is the Von Karman constant, $z$ is negative in the ocean, $u_{*w}$ is the friction velocity in the water
(this being the air friction velocity multiplied by the square root of air to sea density ratio), and the stability
function $\phi_t$ discriminates between a stable and an unstable regime, depending on the sign of its argument:
positive for the stable and negative for the unstable one. Assuming $z$ to be negative in the ocean, the change of
sign entirely depends on the Monin Obukhov length, which is a length characterizing the prevalence of
buoyancy variations induced turbulence over the one generated by wind shear effects. This in turn is strongly
dependent on the sign of the net heat flux $Q$. If $Q>0$, i.e. the ocean gains heat from the atmosphere, and we
have the stable regime: the diffusion coefficients decrease with increasing depth, favoring the downward heat
transfer within the water column. The opposite case, which favors transfer of heat from the ocean to the
atmosphere, can be modeled in different ways (see While et al., 2017 and references therein).
Assuming a temperature of dependence, for $d \gg \delta$ of the form

$$T = T_{-\delta} - \left[\frac{z+\delta}{-d+\delta}\right]^{\nu} (T_{-\delta} - T_{-d}), \quad \nu \text{ empirical parameter} \tag{9}$$

eqn. (7) simplifies to

$$\frac{\partial}{\partial t}(T_{-\delta} - T_{-d}) = \frac{Q + R_s - R(-d)}{d\rho_w c_w}\frac{\nu+1}{\nu} - \frac{(\nu+1)ku_{*w}}{d\phi_t(d/L)}(T_{-\delta} - T_{-d}) \tag{10}$$

In ZB05 scheme (Zeng and Beljaars, 2005), eqs. (5, 10) are the equations for the cool skin and warm layer
respectively. Assumptions on the fraction of solar radiation within this layer and the cool skin depth usually
follow Fairall et al., 1996 parameterization, whose detail are given in the Supplementary Material section.

### 249     3.3    Solar transmission expression

The expression of the solar transmission in Zeng and Beljaars, 2005 is

$$\frac{R(-d)}{R_s} = \sum_{i=1}^{3} a_i e^{-db_i}, \quad \begin{aligned} &(a_1, a_2, a_3) = (0.28, 0.27, 0.45), \\ &(b_1, b_2, b_3) = (71.5, 2.8, 0.07)m^{-1}, \end{aligned}$$

following Soloviev formulation (Soloviev, 1982) (S82 in the following), which is very widely used in
atmosphere models (such as WRF, Skamarock et al., 2019).

So far this is not the only possibility: a formulation with 61 coefficients has been developed by Jerlov, 1968,
which is based on different water types classified based on chlorophyll concentration and particulates, for light
in the visible spectrum.

A formulation with 9 coefficients (reported in Table 2) has been proposed to include such effects (Soloviev and
Schlussel, 1996, Gentemann et al., 2009) the first of them accounting for mean properties of I, IA, IB, II and
III Jerlov's optical water types. This formulation is widely employed in ocean models (such as in the optional





skin SST routine of NEMO, see While et al., 2017), with the reference depth $d$ fixed to 3 $m$. So, the solar
transmission coefficient follows as:

$$\frac{R(-d)}{R_s} = \sum_{i=1}^{9} a_i e^{-db_i} \ .$$

264                                                                       (13)

Ideally, one would like to have a reference depth representative of the one at which the transmission of solar
radiation is negligible, and if we take it as the depth at which transmission drops by 1/$e$ from its surface value,
we get a value which can be different from $d$ = 3 $m$, as we can see from figure 2a. Allowing for a realistic time
and space varying value of $d$ represents the main novelty of our work.

From this viewpoint, choosing a value of $d$ = 3 $m$ while using the solar extinction formulation as in Soloviev,
1982 or Soloviev and Schlussel, 1996 would lead to underestimate the penetration of solar radiation into the
warm layer. Another possibility, as in the case of the NEMO module for radiation calculations (Jerlov, 1968,
Morel et al., 1989, Lengaigne et al., 2007), is to reconstruct a chlorophyll profile from its surface values and
employ an R-G-B scheme to calculate radiation as a function of depth. From eqn. (13) with only 4 terms (one
for chlorophyll, and three for R-G-B), one can numerically derive the e-folding depth using chlorophyll
variations and the R-G-B light extinction coefficients taken from lookup tables in the source code.

This would give a constant transmission throughout the basin, but with a spatially and temporally varying e-
folding depth and defines our new prognostic scheme for skin SST warm layer calculation. Everything else is
left unchanged, both the refinements of Takaya et al., 2010 (T10 hereafter) and the A02 model for cool skin.
**3.3.1    E-folding depth estimates**

282       Mediterranean Chlorophyll climatology data (see section 2.4) were re-gridded onto a 0.25 ° regular

longitude/latitude grid, and tabulated coefficients within NEMO were used to retrieve the transmission,
accounting for chlorophyll variations. E-folding depths then can be estimated as the depth at which transmission
drops by 1/e from its surface value. It can be noticed from figure 2b that also the e-folding depth varies with
seasonality, with typical values ranging from about 3 to 4.5 meters. This is the central point of our modification
to the prognostic scheme.
**3.4    Overview of the simulations performed.**

289       With the coupled ocean-atmosphere regional system we performed a set of four simulations, forced by

ERA5 in the atmosphere and ORAS5 (Zuo et al., 2018) in the ocean and covering three years (from 2019 to
2021), with hourly outputs (a synthesis is provided in Table 1):
1. a control run, in which no skin SST prognostic scheme is activated, therefore the diurnal SST variations

293         in the uppermost ocean layer (0.5 m thick) only come from the variability represented by the ocean model





at about 0.5 m of depth, considering also the 0.5 hours frequency of the coupling. We will refer to this
experiment in the following as *ctrlnoskin*;
2. a run in which the ZB05 scheme in WRF (Zeng and Beljaars, 2005) is active - we shall refer to this case
in the following as *wrfskin*;
3. a run in which the existing scheme within NEMO, which employ the 9-coefficient parameterization for
light extinction coefficients (Gentemann et al., 2009 - G09 hereafter), the scheme for the cool skin as
modified in A02, and refinements of the stability function, in the warm layer formulation as in T10 - we
shall refer to this as the *nemoskwrite* case;
4. a fourth simulation in which we modified the reference depth for the basis of the warm layer from $z = 3$
*m*, to an e-folding depth (i.e. the depth at which radiation gets diminished by $1/e$ from its surface value),
which is allowed to vary temporally and spatially because it is estimated from R-G-B light extinction
coefficients and chlorophyll concentration (see section 3) below. We will refer to it as *modradnemo*,
being the experiment where our modification to the skin SST scheme is implemented and tested.
The reason behind the choice of the above mentioned period of three years 2019-2021 is twofold: firstly, it
allows a validation against all the measurements from different data sources (satellite, drifters and objectively
analyzed profiles), and secondly, it is a good trade-off between the needs of keeping a reasonable computational
load, data volume for the analysis, and guarantees a minimal robustness of our finding, compared to a simulation
which covers just one year. However, we do not discard the possibility to extend the time coverage in our plans
for future works.

## 313  4   Results

In this section, we compare simulations outputs with data from different sources (see section 2), to assess
methods performances and impacts of our modifications. Since we are mainly acting to improve skin SST
diurnal variations reconstruction in the ocean component, the main focus is on the difference between the
nemoskwrite and modradnemo, while the ctrlnoskin and wrfskin ones are included as further reference elements
(the latter being not directly comparable because the atmospheric model sees the ocean foundation SST and
employ the scheme just to diagnose the skinSST).

### 321  4.1   Comparison with CMEMS MED DOISST

We calculated the mean diurnal warming amplitude in each season as the seasonally averaged diurnal
warming amplitudes (diurnal warming amplitude being defined for each day as the difference between daytime
maximum and nighttime minimum of SST), which can be cast into the following equation:

$$\langle \mathrm{DWA} \rangle_{\mathrm{seas}} = \frac{1}{N_{\mathrm{seas}}} \sum_{i=0}^{N_{\mathrm{seas}}} \left\{ \max_{h_i \in [10:00,18:00]} - \min_{h_i \in [00:00,06:00]} \right\} \mathrm{SST}\left(h_i\right),$$

325  (14)



where seas = DJF, JJA, MAM, SON is the given season, $N$seas is the number of days in that particular season
and $h_i$ is the local time in hours for any given day.
Seasonally averaged diurnal warming amplitudes are shown in figure 3. On average, the maximum amplitude
is reached in summer, with the wrfskin simulation peaking at about 3 $K$, thus overestimating the mean diurnal
cycle compared to CMEMS MED DOISST (the monthly biases with respect to CMEMS foundation SST both
in the western and the eastern part of the Mediterranean Sea stay below 1 $K$ year-round for every of the
simulations performed – see figure S1 in Supplementary Materials). The nemoskwrite simulation yields a
pattern very similar to CMEMS MED DOISST in summer, but underestimates the signal in the remaining
seasons. Outside the Summer season, our modifications yield a slight improvement (see modradnemo, last row
of figure 3). As expected, the control run in which no skin SST method is active, generally underestimates the
diurnal signal everywhere. Compared to nemoskwrite, the modradnemo simulation improves JJA mean diurnal
warming amplitude, especially over the Southern Mediterranean Sea, while in central and Northern part of the
basin tends to overestimate the signal by about 0.5 K with respect to CMEMS-DOI data. Furthermore, a general
underestimation is present also in DJF, with the modradnemo simulation showing the smallest differences with
respect to CMEMS-DOI data.
The spatial average over the whole Mediterranean domain is shown in figure 4, confirming the general
underestimation of the control run and the overestimation of the wrfskin (ZB05 scheme) in all seasons except
winter.
Spatially averaging highlights that our modification brings improvement, especially during wintertime, while
in all the other seasons the best agreement is gained by using the nemoskwrite setup (ZB05 with T10 and A02
modifications), at least according to the verification against CMEMS MED DOISST.
On a monthly timescale, figure 5 confirms that the control simulation tends generally to have a negative bias
of the diurnal amplitude, for the whole simulated period. The wrfskin (ZB05 scheme) shows a warm bias during
summertime months, shown just as a reference. The comparison between nemoskwrite (ZB05+A02+T10) and
modradnemo (chl e-folding depth) shows improvement of our scheme (modradnemo) over the old one
(nemoskwrite) especially in May, but not in June, despite in the rest of the period the amplitude of the bias is
slightly reduced.

## 4.2   Comparison with iQuam Star HR-Drifters

The bias with respect to drifter measurements averaged over drifters positions as a function of the month
and time of the day is shown in figure 6. All the schemes present a systematic cool bias in autumn (SON) for
most of the hours of the day. During April and June, the modradnemo simulation significantly reduced the
warm bias with respect to observations, compared to the nemoskwrite case, keeping it however generally





positive. This is quite reasonable, since drifters measurement can be thought representative of a depth which
can be also below the subskin level (typically of the order of some centimeters). Consistently with figure 5, the
wrfskin has a larger positive bias than modradnemo in June.

Further, as shown by figure 7, the bias between CMEMS MED DOISST and drifters is generally positive
anytime except in late spring/summer and autumn during nighttime. This pattern arises because of the
composite effect of having a temperature representative of the subskin level where and when there are data
from radiometers, and a temperature of about 1 $m$ depth from the MEDFS system as first guess of the optimal
interpolation over cloudy regions (Pisano et al., 2022). However, the modified scheme significantly reduces the
difference, yielding a bias closer to the one of CMEMS MED DOISST with respect to drifters, especially during
April, which is the month in which the number of observations from drifters is definitely larger.

### 4.3    Comparison with EN4 objective analysis

Bias corrected vertical profiles gathered in an objective analysis were used to assess differences across
schemes along the water column. To summarize we report here only a macro subdivision into the eastern and
the western Mediterranean Sea, respectively in figures 8, 9. Model outputs were remapped on the same vertical
and horizontal grid. Looking at the mean profile averaged over all grid points in the given area, the agreement
is better for all simulations during summertime months, both for the eastern and the western region (see figs.
8c, 9c), showing in particular that the modradnemo simulation outperforms the nemoskwrite one. This is also
true for the wintertime season in the eastern Mediterranean (see fig.8b). On the other hand, in the western
Mediterranean all simulations tend to overestimate the signal, with our modified scheme doing a better job.
However, below about 80 $m$ depth differences across schemes vanish.

Looking in more detail at the RMSE on the top 15 $m$ depth between each simulation and EN4 as a function of
the month and more detailed region subdomains shown in figure 10a, we can see how in general all simulations
present the same pattern for the region outside of Gibraltar Strait, which can be thought an effect related to the
presence of the relaxation to horizontal boundary conditions, while for all the remaining regions and months
the control run, the wrfskin and the modradnemo present a similar pattern, with the modradnemo reducing the
RMSE in most of the regions and for most of the months, especially with respect to nemoskwrite, and this is
particularly true over the central Mediterranean Sea, in regions like Thyrrenian and Adriatic Seas.

### 4.4    Heat fluxes and vertical propagation

In this section we aim to characterize the differences of each scheme with respect to the control simulation.
We do this by specifically looking at the seasonality of Mixed Layer Depth (MLD), vertical profiles of
temperature in specific months and regions, and via the comparison of the net surface heat fluxes over the whole
Mediterranean Sea.





Compared to the Mixed Layer Depth climatology from 1969 to 2013 (Houpert et al., 2015a, Houpert et al.,
2015b, section 2.7), all of the tested schemes seems to have a similar impact on Mixed Layer Depth's
seasonality, with larger differences with respect climatological values being mostly located in the Eastern
Mediterranean Sea and during wintertime/spring (Figure 11). Figure 12 show how our modified scheme allows
more (less) vertical propagation of the diurnal signal during summer (winter) with respect to schemes with
constant e-folding depth in all central regions of the Mediterranean domain (regions 2, 3, 4 as defined in figure
10a), when all of them are referenced to the control simulation temperature daily minimum.
Indeed, from figure 12b, we can see that when all the temperature profiles for each simulation are referenced
to the ctrlnoskin daily minimum, there is a much wider diurnal warming signal for most of all the considered
depths level, with modradnemo representing an intermediate situation between the wrfskin and the nemoskwrite
simulation. This is probably due to the inclusion of chlorophyll-interactive variations, which allow for a better
representation of the variability of the mixed layer dynamics.
Estimates of the mean Mediterranean heat exchange between ocean and atmosphere based on previous studies
range from -11 to +22 W/m$^2$, with an evident dominance of negative estimates, i.e., heat loss from the ocean to
the atmosphere (Jordà et al., 2017, Pettenuzzo et al., 2010). Some other studies suggest that the Mediterranean
heat budget is close to a neutral value, -1 W/m$^2$ (Ruiz et al., 2008) or +1 W/m$^2$ (Criado et. al., 2012). Many
factors can contribute to such wide variability among different estimates, such as differences in the
parameterizations employed, initial and boundary conditions, and the way the physical processes, especially
through the Strait of Gibraltar are modeled (Macdonalds et al., 1994, Gonzales, 2023).

As shown by table 3, all simulations on an annual basis give a negative, non-closed balance for the net surface
heat flux, and modifications to include skinSST, performing very similarly one to another, bring the budget by
1.5 $W/m^2$ closer to zero, while ERA5 data show a positive net surface heat flux close to 5 $W/m^2$. However, all
estimates fall into the (large uncertain) literature-based estimates. On seasonal timescales, the inclusion of
skinSST diurnal variations has the following effects:
• less net heat loss to the atmosphere during wintertime with respect to the control run (wrfskin differing
from the ctrlnoskin by about 6$W/m^2$, while nemoskwrite and modradnemo having a similar impact, with
a difference of about 4$W/m^2$ with respect to the control run);

• in springtime, all simulations show a positive imbalance, with the highest difference with respect to the
control run of about 1 $W/m^2$ in the modradnemo simulation;
• during summer, our modified scheme brings on average about 3 $W/m^2$ more than the control simulation
into the basin, yielding an estimate which is closer to ERA5;
• in autumn, our scheme cools down more than the control (about 2 $W/m^2$), being the farthest simulation
from ERA5 estimate, while traditional schemes tend to have a less negative net heat input.

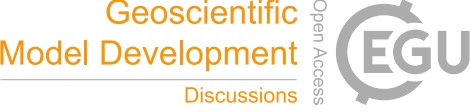

All seasons except spring show larger difference with respect to ERA5 fluxes, with underestimation in summer,
and overestimation during winter and autumn, resulting in a bias of about 10 $W/m^2$ with respect to the net heat
flux annual budget in ERA5.
## 5    Summary and Conclusions
In this paper we studied the sensitivity of a regional coupled ocean-atmosphere-hydrological discharge
regional model on the Mediterranean Sea to prognostic schemes for skin sea surface temperature. Specifically,
we developed a new scheme which allows for spatial and temporal variations of the warm layer's extent
according to seawater's transparency conditions. This is possible by using tabulated solar extinction coefficients
already used in the ocean model, and inverting the functional form which determines how the solar radiation
varies along the vertical direction to find the depth at which this latter drops by $1/e$ from its surface value.

We simulated the period 2019-2021, analyzing hourly model outputs, and comparing aggregated results with
satellite, objectively analyzed and drifters data. Overall, the comparison with data shows that the new scheme
improves what is already implemented in NEMO, e.g. mean diurnal warming amplitudes are closer to satellite
observations in winter, spring and autumn, not being much worse than other existing schemes in summer, at
least looking at maps of mean diurnal warming amplitude grouped by seasons. Looking to the typical
temperature profile in both the eastern and the western Mediterranean Sea, non-negligible differences across
schemes stay confined in the topmost $20m$ ($100m$) of depth during summertime (wintertime). Regionally,
typical profiles are warmer than EN4 observation year-round for western regions (regions -1,1,2) especially in
winter, while regions in the east show a smaller RMSE in the topmost meters for basically all the regions and
months when comparing modradnemo to nemoskwrite. The Adriatic Sea has a systematically higher RMSE
with respect to EN4 in all the tested methods, for the whole period considered. In the central regions, the new
scheme penetrates temperature anomalies more (less) during summer (winter) months, having a less intense
mean diurnal warming amplitude signal in summer, especially over the upper few meters (the converse holds
for wintertime values). Therefore, with respect to the ctrlnoskin simulation, nemoskwrite shows the coldest
signal, the wrskin the hottest, and our modification modradnemo constitutes the middle situation, with milder
summer and winter than the control run. Therefore, future research efforts should be devoted to the better
characterization of this aspect, especially to understand if the modified vertical penetration of heat has some
particular effect on the dynamics of the mixed layer (see Song and Yu, 2017 and references therein). On a long-
term perspective, the method needs to be tested also in other areas and for longer periods, which can increase
the results' certainty and allow for usage in investigating impacts on relevant climate large-scale phenomena,
where the role of an improved diurnal warming signal could be more relevant (Bernie et al., 2007, Bernie et al.,
2008). These includes phenomena and physical processes such as propagation of Marine Heat Waves (MHW)
or deep water formation and deep convective events.




*Code and data availability*

The NEMO ocean model code (v4.0.7) is available at https://forge.ipsl.jussieu.fr/nemo/wiki.

The WRF atmospheric model code (v4.3.3) is available at https://github.com/wrf-model/WRF.

The HD hydrological discharge model (v5.1) is available at https://zenodo.org/record/5707587#.Y-0VQ3bMKUk.

The frozen version of the MESMARv1 code used in this manuscript is available at: https://doi.org/10.5281/zenodo.7898938.

CMEMS MED DOISST Data downloaded from CMEMS portal.

Chlorophyll data are freely available from CMEMS portal.

The iQuam data version of this study used is V2.1, downloaded from the National Environmental Satellite, Data, and Information Service Satellite Applications and Research NOAA NESDIS STAR portal.

Gridded analyses of EN4 profiles are distributed from the MetOffice Hadley Centre Observations (we used version 4.2.1).

ERA5 data are freely available after registration on the Climate Data Store (CDS) by Copernicus Climate Change Service (C3S).

MLD data are distributed on a 0.25 degree regular grid, and freely available from the Sea Open Scientific Data Publication SEANOE portal.

Minimal data and scripts used within the manuscript to reproduce the figures in the manuscript are available at this link:
https://zenodo.org/records/10451206



*Acknowledgments.* We specifically acknowledge Olivier Marti (LSCE/IPSL) and Aurore Voldoire (CNRM-
CNRS) for fruitful discussion during the 6th Workshop on Coupling Technologies for ESM held from 18 to 20
January 2023 in Tolouse, and Sophie Valcke (CERFACS) for their valuable support in the use of the NEMO
and WRF interfaces to the OASIS coupler. This Research is supported by ICSC – National Research Center for
High Performance Computing, Big Data and Quantum Computing, funded by Ministero dell'Istruzione,
dell'Università e della Ricerca through the NextGenerationEU programme.

*Author Contributions.* VdT and AS conceived the study and designed the experiments to conduct, VdT
Performed the simulation and data analysis, data downloading and writing of the first draft, VdT, DC, YH, CY,
VA, AP, DC, RS and AS equally contributed to discuss and interpret the results, finalizing the draft.

*Competing Interests.* All authors declare they have no competing interests.



# Figures

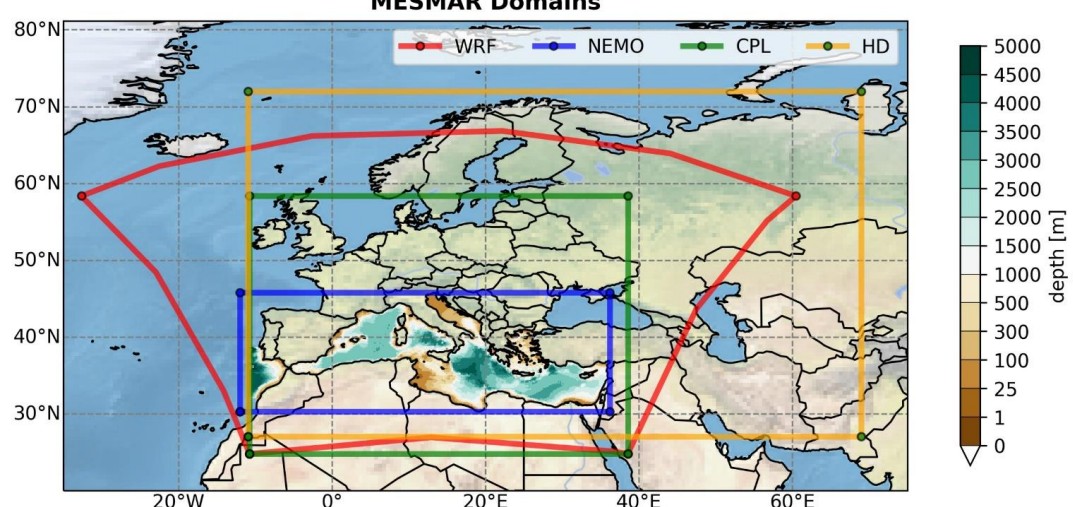

**Figure 1**: The modeling system domain: WRF, NEMO, HD and boundaries for the coupling mask are respectively in red, blue, orange, and green. Contour filled plot shows the ocean model bathymetry.



515

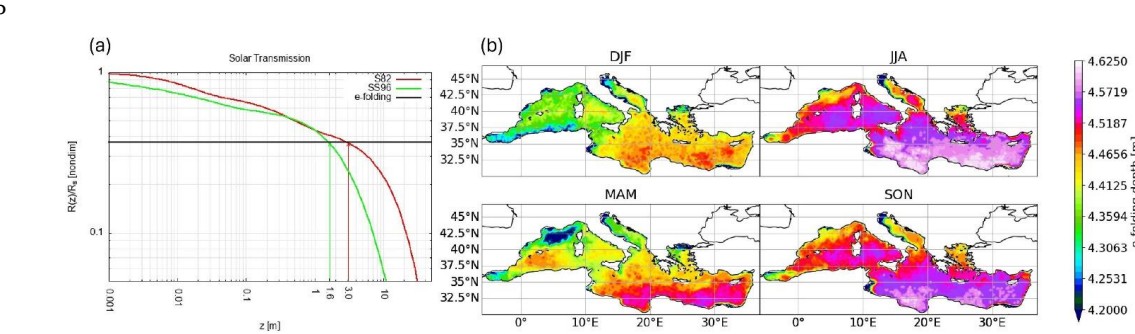

516

**Figure 2**: Panel 2a shows two different formulations frequently used for the transmission coefficient expression: the red curve shows the formulation of Soloviev, 1982, while the green curve the one defined in Soloviev and Schlussel, 1996. Panel 2b shows e-folding depth estimates from Mediterranean Chlorophyll climatology of Volpe et al., 2019: lowest values touch the 2.5 meters.






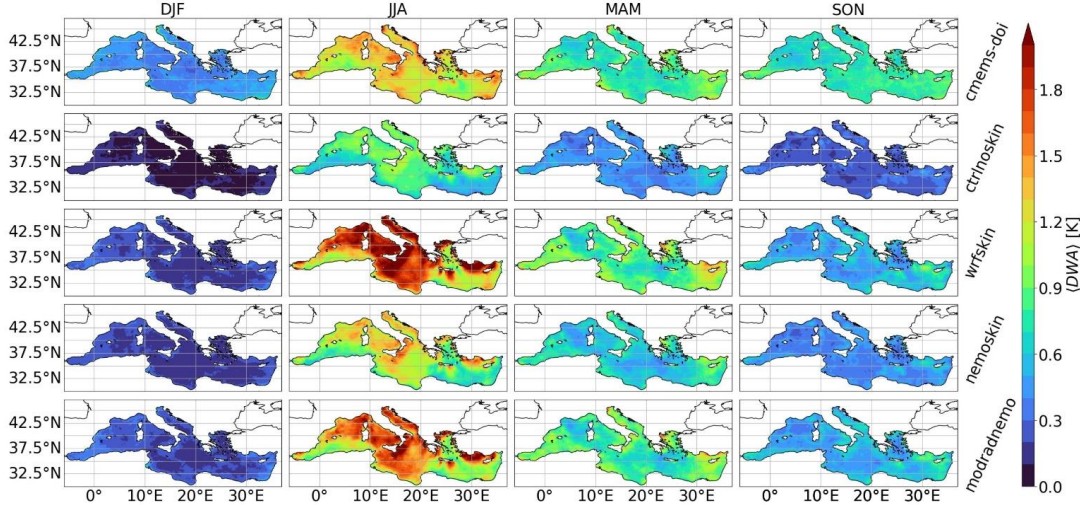


**Figure 3**: Mean diurnal warming amplitude averaged over seasons (on columns), for each case (row): the first
row is the CMEMS MED DOISST data, followed in order by the control simulation, wrfskin, nemoskwrite and
modradnemo.







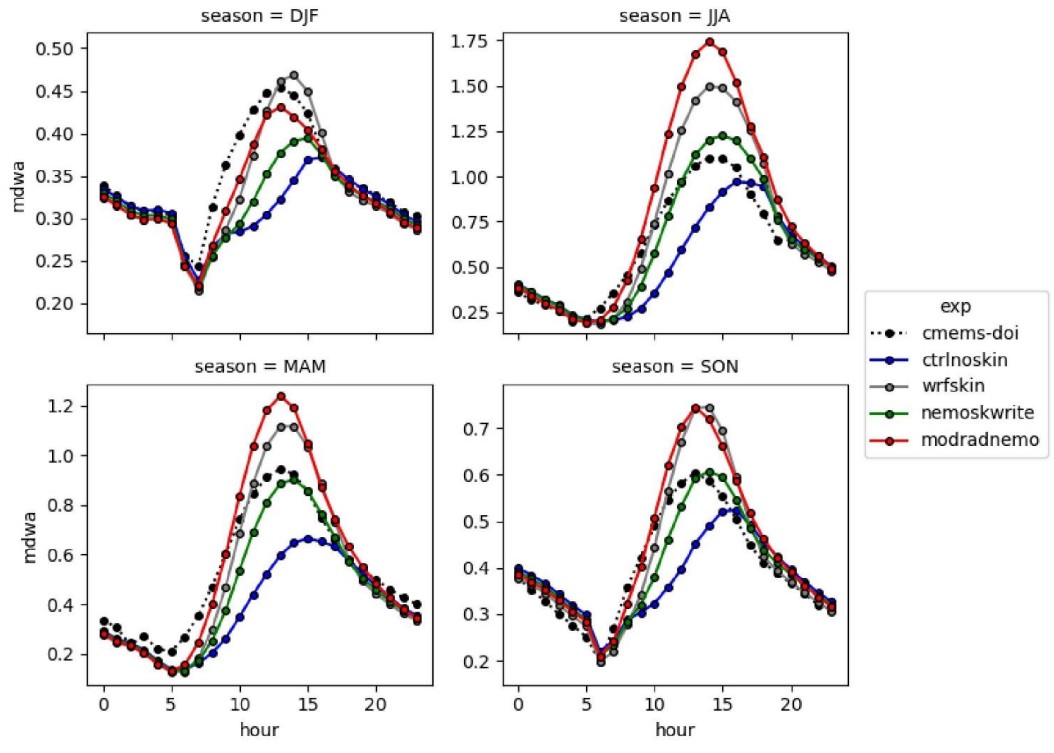


**Figure 4**: Seasonality of the diurnal cycle averaged over the whole Mediterranean Sea, masking out regions in time and space where the percentage of model data in CMEMSDOI is greater than 50%.


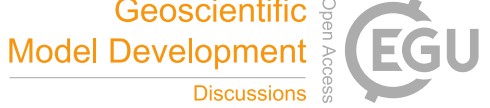


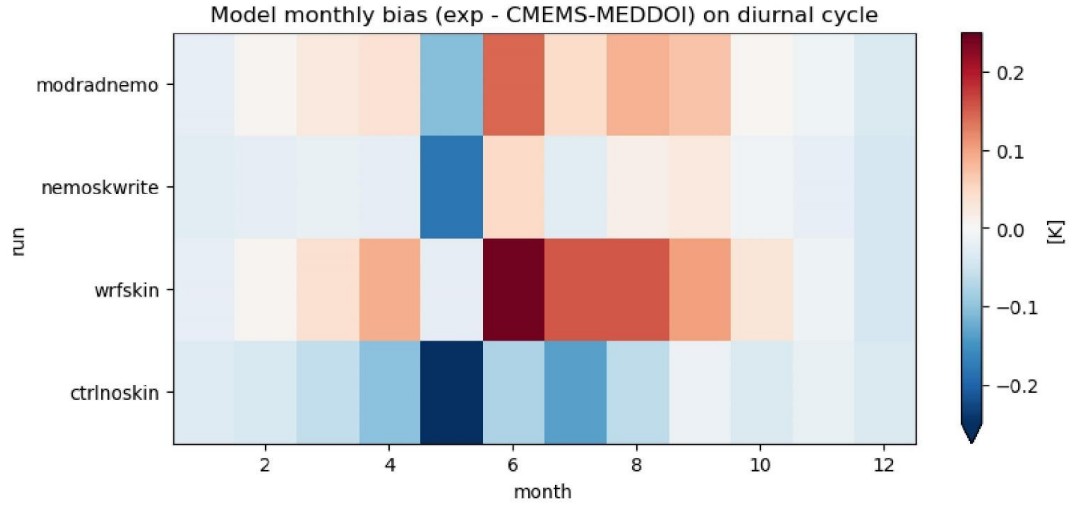


**Figure 5**: Monthly averaged values for the time series of spatial mean diurnal cycle over the Mediterranean Sea
(bias with respect to CMEMS MED DOISST)


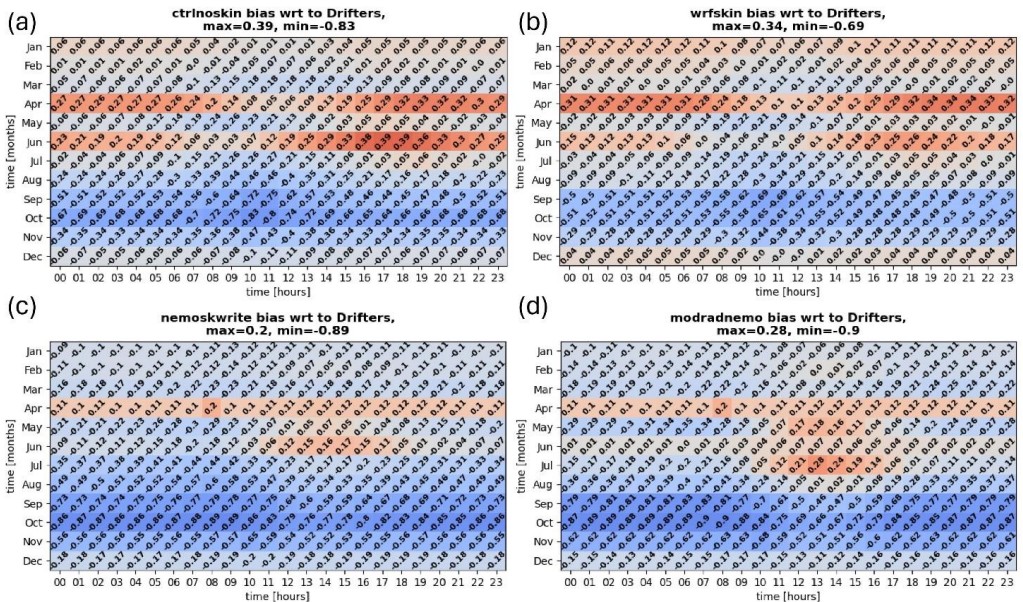

**Figure 6**: Bias with respect to measurements averaged over drifters' locations as a function of the month and the time of the day. Panels 6a, 6b, 6c, 6d show respectively the results for all the simulations carried out in the present study. Confidence on these numbers can be supported by the numbers of measurements reported in table S1.






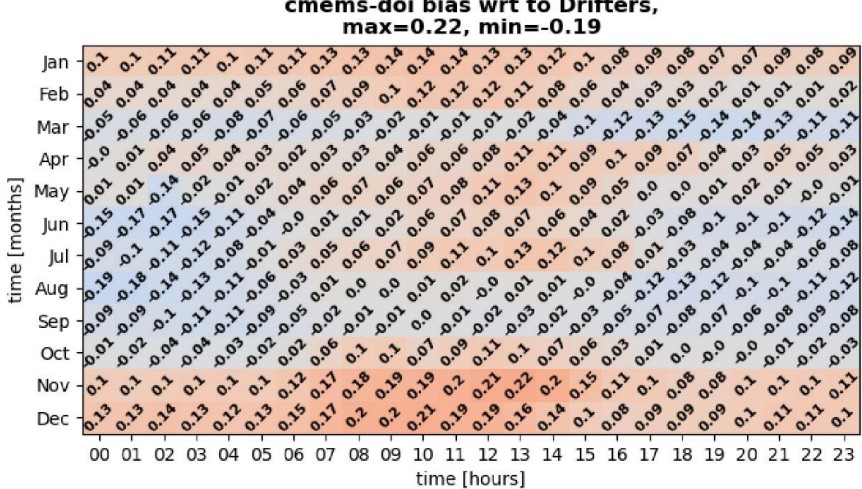


**Figure 7**: Bias with respect to measurements averaged over drifters' locations as a function of the month and time of the day, for CMEMS MED DOISST data.





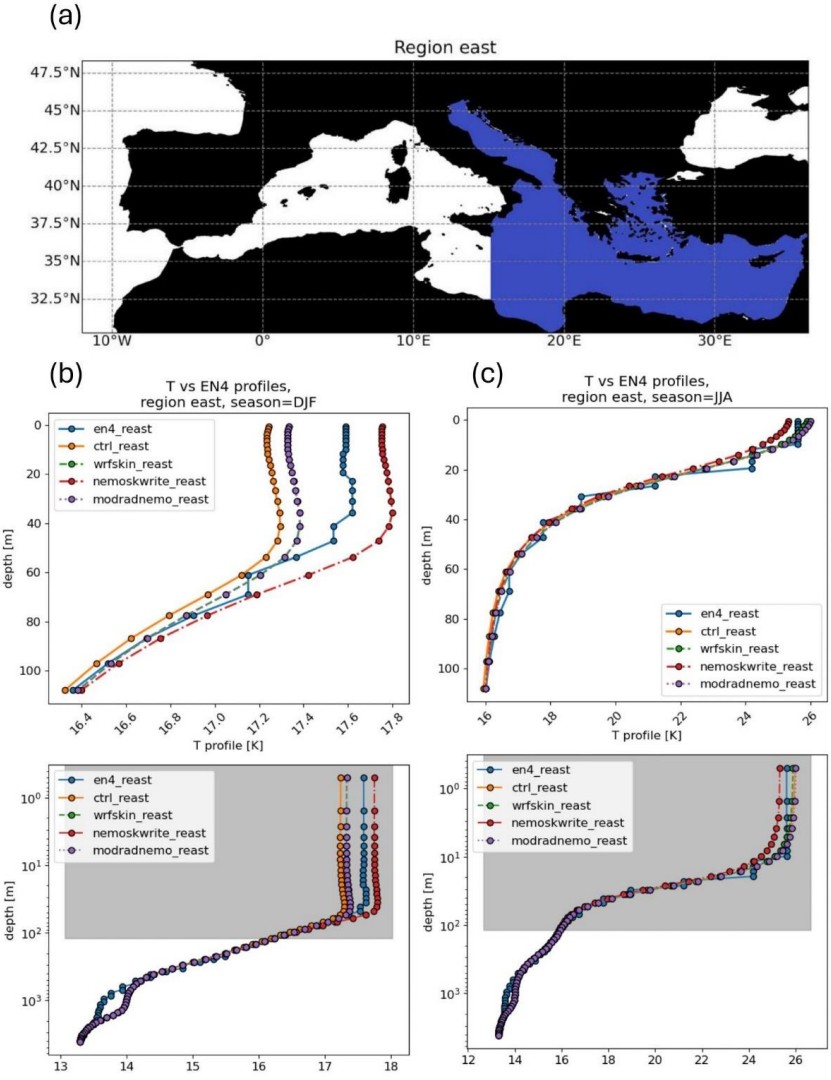

**Figure 8**: Spatial average of profiles within the eastern Mediterranean Sea, during winter and summer. Panel
8a shows the eastern region, while 8b, 8c show respectively wintertime and summertime spatially averaged
profiles within the top 100 *m* in the upper part, on the bottom the whole depth range on a logarithmic scale.


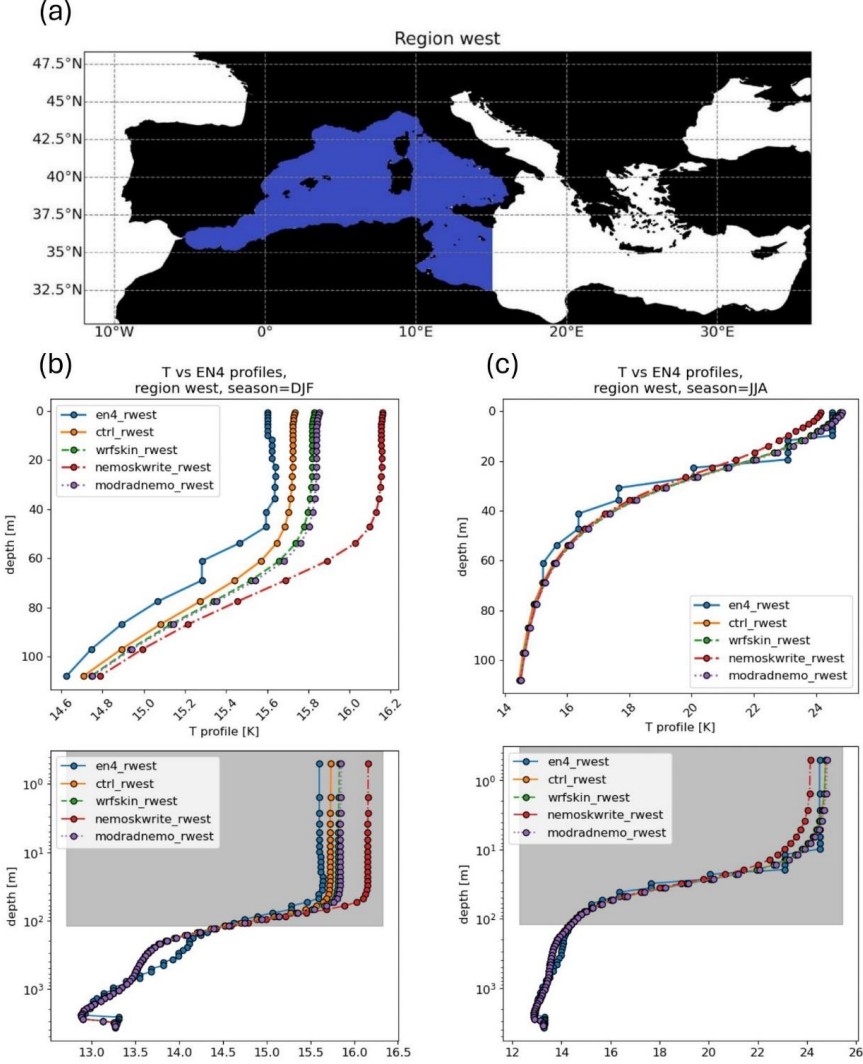

**Figure 9**: Spatial average of profiles within the eastern Mediterranean Sea, during winter and summer. Panel
9a shows the eastern region, while 9b, 9c show respectively wintertime and summertime spatially-averaged
profiles within the top 100 *m* in the upper part, on the bottom the whole depth range on a logarithmic scale.



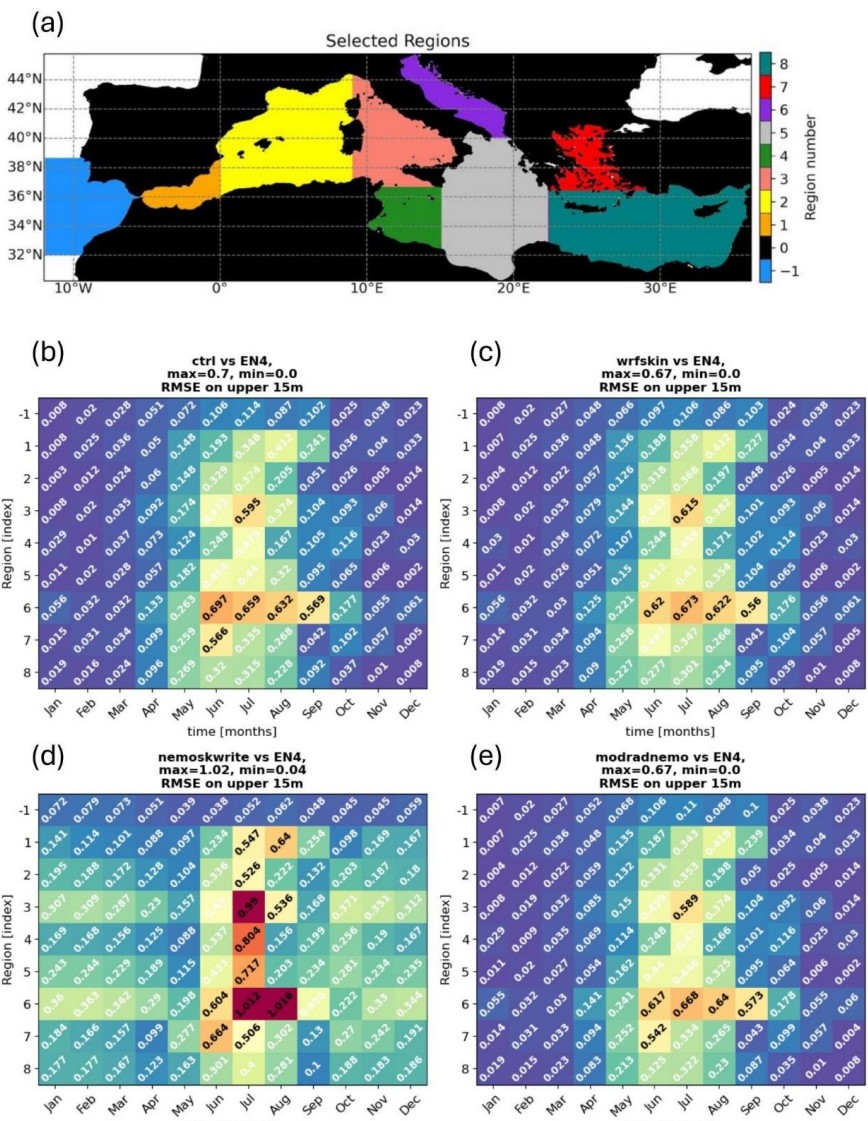

**Figure 10**: RMSE on the top 15m of the difference between regionally averaged profiles between each
simulation and EN4, displayed as a function of the region and the particular month. Division in regions is
reported in panel 10a, while 10b, 10c, 10d, 10e show respectively the results for all the simulations carried out
in the present study.



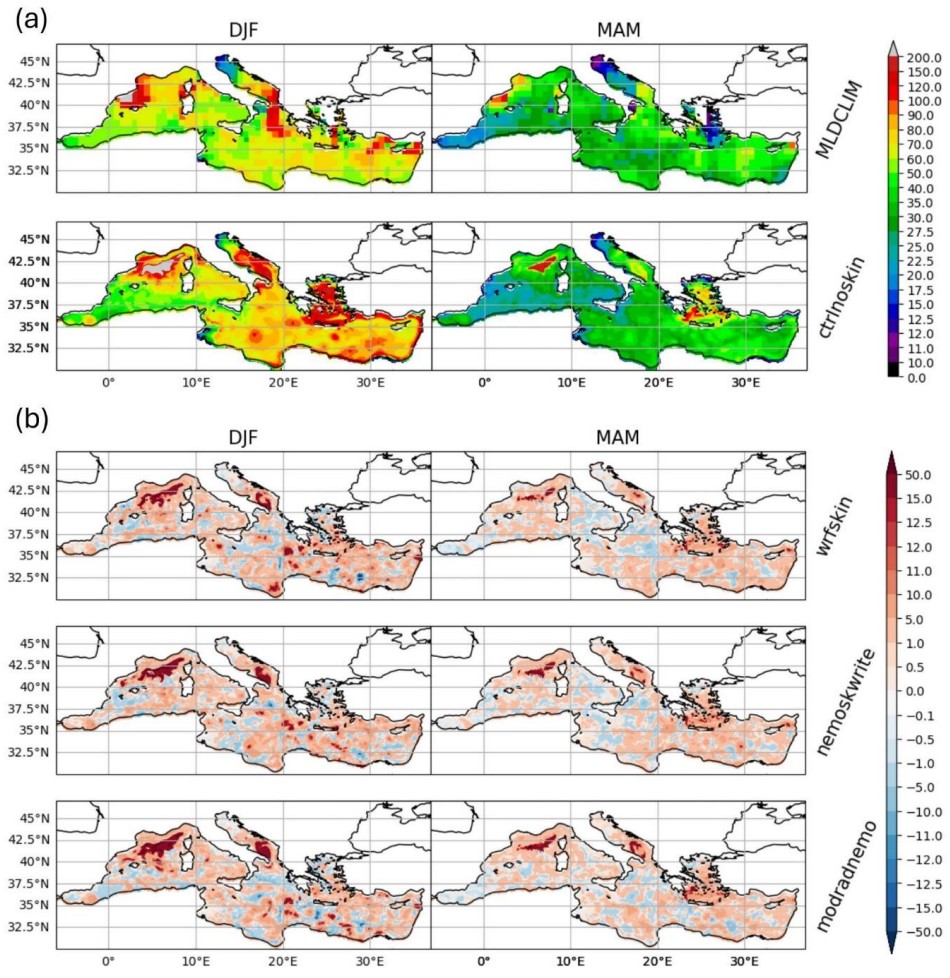

**Figure 11**: Maps of DJF, MAM of mixed layer depth for the climatology and for the control simulation in panel (a). Panel (b) shows the difference of the control with respect to each simulation. Units are meters.




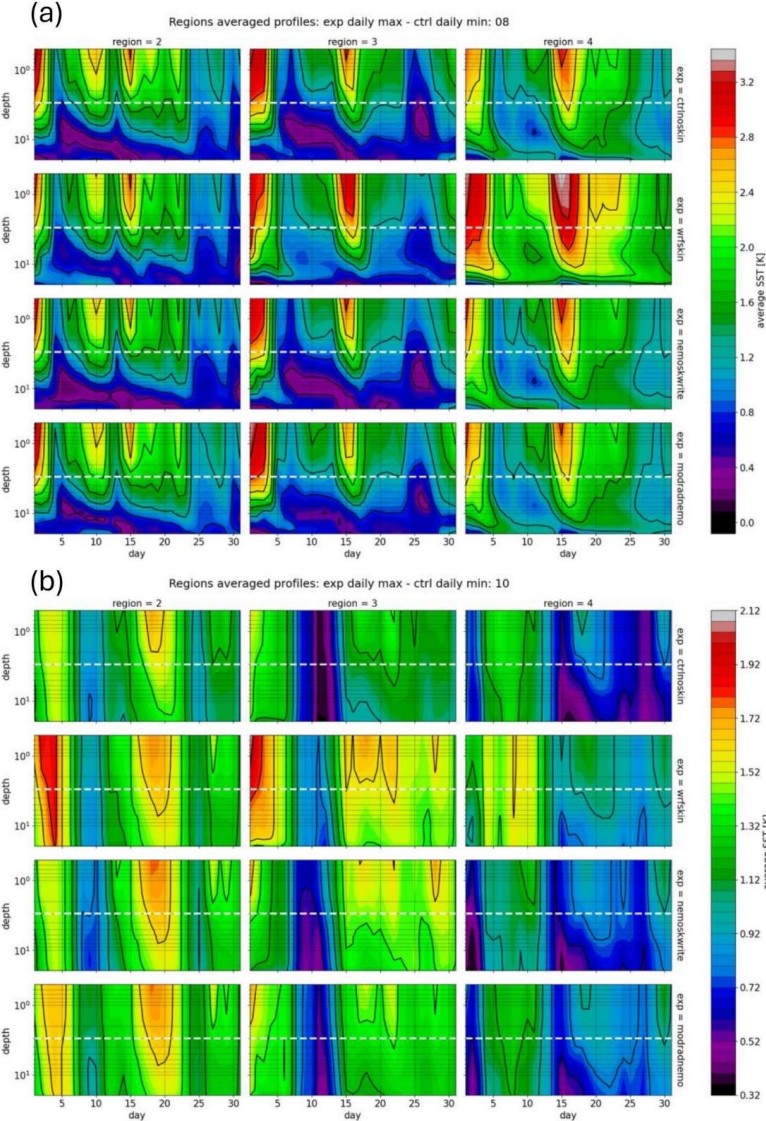


**Figure 12**: Hovmoller plots for spatial average of model outputs temperature profiles in the regions 2,3,4 as
defined by figure 10a. Each row shows the difference between daily maxima for the given experiment minus
the daily minima for the control simulation. The white dashed line traces the $z = 3m$ line of the depth used as
reference for the base of the warm layer as in ZB05 scheme Zeng and Beljaars, 2005. Panel 12a shows August,
panel 12b shows October.



Tables

| Simulation | Scheme active | Extinction coefficients in Warm Layer |
|---|---|---|
| ctrlnoskin | None | None |
| wrfskin | ZB05 | SS82 |
| nemoskwrite | ZB05+A02+T10 | G09 |
| modradnemo | ZB05+A02+T10 | R-G-B + chl e-folding |

**Table 1**: Overview of the simulations performed

| Wavelength [$\mu m$] | i | $a_i$ | $b_i\ \left[m^{-1}\right]$ |
|---|---|---|---|
| 0.3-0.6 | 1 | 0.2370 | $1.488 \times 10^{-1}$ |
| 0.6-0.9 | 2 | 0.3600 | $4.405 \times 10^{-1}$ |
| 0.9-1.2 | 3 | 0.1790 | $3.175 \times 10^{1}$ |
| 1.2-1.5 | 4 | 0.0870 | $1.825 \times 10^{2}$ |
| 1.5-1.8 | 5 | 0.0800 | $1.201 \times 10^{3}$ |
| 1.8-2.1 | 6 | 0.0246 | $7.937 \times 10^{3}$ |
| 2.1-2.4 | 7 | 0.0250 | $3.195 \times 10^{3}$ |
| 2.4-2.7 | 8 | 0.0070 | $1.279 \times 10^{4}$ |
| 2.7-3.0 | 9 | 0.0004 | $6.944 \times 10^{4}$ |

**Table 2**: Parameters for the Transmission coefficient following Soloviev and Schlüssel, 1996.





| simulation | DJF | MAM | JJA | SON | Annual |
|---|---|---|---|---|---|
| ctrlnoskin | -173.31 | 133.92 | 75.56 | -66.40 | -7.55 |
| wrfskin | -168.83 | 134.19 | 76.51 | -65.87 | -5.97 |
| nemoskwrite | -169.28 | 133.79 | 76.77 | -65.72 | -6.10 |
| modradnemo | -169.06 | 134.87 | 78.16 | -68.13 | -6.04 |
| ERA5 | -140.36 | 133.24 | 81.96 | -53.46 | 5.35 |

**Table 3**: Averaged surface net heat flux over the Mediterranean Sea ($W/m^2$): seasonal and annual spatial averaged mean values.



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
