# Peer review of "Skin Sea Surface Temperature schemes in coupled ocean atmosphere modeling: the impact of chlorophyll-interactive e-folding depth."

_Geoscientific Model Development, 2024_

## Author Comment (AC1)

**Response to Reviewer 1 comments**

We wish to thank Reviewer 1 for dedicating time to carefully read our work and providing his/her feedback. We sincerely think his detailed comments helped us to improve the manuscript. Here it follows a point-by-point response to the reviewer's report (text in bold denotes the provided comments, while normal text denotes our response), associating with the revised manuscript with the track of the changes.

**This paper examines the ocean radiant heating parameterization and its impact on prognostic schemes simulating the diurnal cycle of SST. Typically, in such prognostic schemes (e.g., Zeng & Beljaars (2005)) the diurnal warm layer depth is fixed at 3m. This assumes negligible diurnal warming and minimal radiative heating below this depth; however, this paper notes that this is not necessarily the case and suggests that the warm layer depth in the prognostic scheme should instead vary spatially and temporally depending on the e-folding depth derived using chlorophyll data. This seems to me to be a worthy improvement and an interesting study.**

**The default parameterization for solar radiative heating in such schemes uses a 9-band decomposition of the solar radiation with invariant coefficients. However, it is well known that water properties can influence solar penetration; hence, the development of parameterizations with improved skill that make use of ocean color data. For example, Morel & Antoine (1994), Ohlmann & Siegel (2000), and Lee et al. (2005). These types of schemes are not mentioned in the manuscript, but I think provide helpful background. How does the new scheme compare to these and why not implement one or more of the above schemes for comparison?**

Thanks for pointing out these additional references. They are very useful to enrich the introduction of our paper, providing more detailed context. In the present work, we decided to restrict the comparison with the ones already implemented within NEMO and WRF. Implementing and testing additional schemes requires an effort in terms of time which goes beyond the one given for this revision, but is a very good hint for future work.
We added a paragraph in the introduction mentioning these works:
"Within these prognostic schemes, seawater's transparency conditions (e.g., estimated using chlorophyll concentration) have great implications in the way solar radiation is absorbed within the ocean's uppermost layer (Morel and Antoine 1994). Ohlmann et al. 2000 quantified with the help of radiative transfer calculations effects of physical and biological processes on solar radiation transmission, classifying as main factors chlorophyll concentration, cloud cover and solar zenith angle. Ohlmann and Siegel 2000 and Lee et al. 2005 are further examples of how radiative transfer models are used to develop solar transmission parameterization which is fit to the sum of exponentials (the number of terms in the sum depending on the variable which has been considered). To the best of our knowledge, these ideas have not been implemented nor tested within the prognostic scheme for skin SST present in the ocean model NEMO, which just relies on chlorophyll-calibrated coefficients though (Gentemann et al 2009).
Our main aim here is therefore to improve existing skin SST prognostic schemes, investigating the impact of variable seawater's transparency conditions ) in modeling solar

radiation extinction in the upper ocean. The use of chlorophyll concentration as a proxy for seawater's transparency is not new. In fact, given its covariance with Secchi disk depth (estimated from reflectance at various wavelength), it has been often applied by the ocean color community to study the dynamics of oligotrophic gyres (Leonelli et al., 2022 and references therein)."
See LL 98-113 of the file with the track of the changes.

**This study uses code from established and widely-accessible models.**

**Specific comments and technical corrections:**

**Line 19-20: Clarify this sentence – as the new scheme also truncates solar transmission! The difference seems to be between a fixed warm layer depth vs a spatially and seasonally adjusted warm layer depth.**

That's right, and to improve the readability, we rephrased the sentence from
"All existing methods truncate solar transmission coefficient at a constant warm layer reference depth; instead, we develop a new scheme where this latter is estimated from a chlorophyll dataset as the e-folding depth of solar transmission.This allows spatial and temporal variations of the warm layer extent to depend on seawater transparency."
to
"All existing methods truncate solar transmission coefficient at a warm layer reference depth which is constant in space and time; instead, we develop a new scheme where this latter is estimated from a chlorophyll dataset as the e-folding depth of solar transmission, therefore allowing it to vary in space and time depending on seawater's transparency conditions."
We hope this clarifies a bit.
See LL 19-22 of the file with the track of the changes.

**Line 23: "… improves the diurnal signal …" compared to what? When you refer to the "old one" which old one as there are 2!? It seems to me that modradnemo is clearly better than wrfskin, but similar (or even worse) than nemoskwrite. What is the evidence/argument from the results that modradnemo is better than nemoskwrite?**

This sentence meant to refer to the improvement of the new method with respect to the one already implemented within NEMO scheme. We have adopted the sentence in the revised manuscript accordingly to provide more clear and a precise description, highlighting the fact that these conclusions function in both time and place.
We rephrased the sentence from
"Comparison against satellite data shows that our new scheme improves the diurnal signal especially during winter, spring, and autumn, with an averaged bias on monthly scales year-round smaller than 0.1 K."
to
"Comparison against satellite data shows that our new scheme, compared to the one already implemented within the ocean model, improves the spatially averaged diurnal signal, especially during winter, and the seasonally averaged one in spring, and autumn, while showing a monthly, basin-wide averaged bias smaller than 0.1 K year-round."
See LL 23-27 of the file with the track of the changes.

Added also a further specification of areas and times where the new scheme improves the old one looking at the RMSE with respect to EN4 objectively analyzed profiles:
"More in detail, the new scheme reduces the RMSE on the top 15 m in the central Mediterranean for summertime months, compared to the one already implemented within the ocean model."
See LL 32-34 of the file with the track of the changes.

**Abstract: "… we develop a new scheme …" "Our "chlorophyll-interactive" method". Can you clarify, what is the actual new contribution here? Is it to try out (implement) an already available method within NEMO (as I understand from lines 270-276) within a prognostic scheme for skin SST?**

Our contribution is to take the existing method within NEMO and modify it to have, instead of a constant warm layer depth and a solar transmission with basin-wide variations, a warm layer reference depth whose spatiotemporal variations are built in such a way that the solar transmission stays constant throughout the basin. This is reached via the inclusion in the RGB light extinction coefficients of the Chl profiles, calculation of the solar extinction profile and retrieval of the warm layer's reference depth as the one at which this solar extinction profile drops by $1/e$ from its surface value.
We changed the wording in lines 270-276 of the original manuscript from
"From this viewpoint, choosing a value of $d = 3$ m while using the solar extinction formulation as in Soloviev, 1982 or Soloviev and Schlussel, 1996 would lead to underestimate the penetration of solar radiation into the warm layer. Another possibility, as in the case of the NEMO module for radiation calculations (Jerlov, 1968, Morel et al., 1989, Lengaigne et al., 2007), is to reconstruct a chlorophyll profile from its surface values and employ an R-G-B scheme to calculate radiation as a function of depth. From eqn. (13) with only 4 terms (one for chlorophyll, and three for R-G-B), one can numerically derive the e-folding depth using chlorophyll variations and the R-G-B light extinction coefficients taken from lookup tables in the source code."
to
"From this viewpoint, choosing a value of $d = 3$ m while using the solar extinction formulation as in Soloviev, 1982 or Soloviev and Schlussel, 1996 would lead to underestimating the penetration of solar radiation into the warm layer. Another possibility, which constitutes our modification to the scheme already implemented in NEMO, is to reconstruct a chlorophyll profile from its surface values following what is already implemented in the NEMO module for radiation calculations (Jerlov, 1968, Morel et al., 1989, Lengaigne et al., 2007), and employ an R-G-B+Chl-a scheme to calculate radiation as a function of depth. Then, from eqn. (13) with only 4 terms (one for chlorophyll, and three for R-G-B, expressed in lookup tables), one can numerically derive the warm layer reference depth as the e-folding depth of the light extinction profile."
We hope this clarifies.
See LL 301-310 of the file with the track of the changes.

**Line 67: suggest rewording**

Accepted. Rephrased from
"Due to its interactions with the atmosphere, the temperature right at the ocean surface is supposed to be almost anywhere and anytime cooler than the ones below, resulting in the

ocean being covered with a cool skin layer: one of the very first and simpler models assumes this cool skin temperature difference as proportional to the ratio between heat fluxes and kinematic stress (Saunders, 1967), via the Saunders' constant."
to
"Due to its interactions with the atmosphere, the temperature right at the interface of separation is supposed to be almost anywhere and anytime lower than the temperature of the waters infinitesimally close to it, resulting in the ocean being covered with a thin cool skin layer. One of the very first and simpler models assumes this cool skin temperature difference as proportional to the ratio between heat fluxes and kinematic stress (Saunders, 1967), via the Saunders' constant."
See LL 71-76 of the file with the track of the changes.

**Line 77: should be 2021 not 2020**

Thanks. Changed from "Zhang et al. 2020" to "Zhang et al. 2021".
See LL 83 of the file with the track of the changes.

**Line 79: "… by most of existing …" --> "… by most existing …"**

Thanks. Fixed Change from "... by most of existing …" to "... by most existing …".
See LL 85 of the file with the track of the changes.

**Line 100-101: suggest rewording final sentence: "We redirect the documentation …"?**

Thanks, done. Rephrased from
"We redirect the documentation and the appropriate literature describing each data and model in depth."
to
"We refer readers to the documentation and relevant literature for detailed information on each dataset and model."
See LL 120-121 of the file with the track of the changes.

**Line 125: I don't think "resumes" (and "resuming" in the table caption) are the correct or best word choice, perhaps "outlines" (and "outlining").**

Done, thanks. changed "resumes" with "outlines" and "resuming" with "outlining".
See LL 150 of the file with the track of the changes and the supplemental material (Table 1 caption).

**Line 149: remove "at day" (or possibly replace with "today")!?**

Thanks, We considered the suggestion in the revised version.
See LL 174 of the file with the track of the changes.

**Line 191: Do you consider sea surface albedo? Is R_s net; i.e., after albedo reflection?**

Yes, we considered it. For more detailed information about the setup, see the NEMO documentation (cited in the manuscript), and the paper by Storto et al., 2023 presenting the MESMAR model, with that, there's also the zenodo link to the complete software.

We added a mention, modifying the sentence from

"Where R_s is solar radiation at the surface"

to

"Where R_s is the net solar radiation at the surface (constant, open-ocean albedo, since the Mediterranean Sea is an ice-free basin)"

See LL 216-217

**Line 205: reword "(already known from decades at those times)"**

We eliminated the sentence.

See LL 231 of the file with the track of the changes.

**Line 215: "neglecting the effect of solar radiation". Saunders (1967) did in fact recognize solar radiation within the skin layer, see their equation (6). It's probably least confusing to just replace Q in your equation (6) with Q + f_s*R_s where f_s is given by equation ? (currently in supplemental materials).**

That's exactly the problem, f_s cannot be given by the expression in the supplemental materials, because that expression came only after Saunders' paper. Indeed, the equation (6) refers to equation (3) of Saunders. We added a sentence to highlight the discussion of Saunders about the inclusion of the effect of solar radiation: "... neglecting the effect of solar radiation (which however recognized its role and added a discussion on how to account for it in the model only at the end of his paper)."

See LL 241-242 of the file with the track of the changes.

**Lines 215-216: This sentence can be improved.**

We added a few words to clarify "...becomes problematic in limiting cases of low and very high wind speeds (greater than 7 m/s), because the wind stress in the denominator limits its validity."

See LL 243-244 of the file with the track of the changes.

**Note that the limiting case of low wind was noticed before Artale; e.g., Fairall et al. (1996) had already corrected for this. I think it is helpful to provide all the equations used in the code (e.g., include the Artale expression for lambda), perhaps in the supplemental materials.**

Thanks for the suggestions. To avoid confusing the reader with too many additional details we preferred to keep the focus on the parts we have worked with, referring to the Zenodo repo (in which we added the nemo routines we modified) if one is interested in having more detail. We added citations to the work of Fairall et al. 1996.

See LL 242.

**I encourage the authors to thoroughly check the equations to make sure they are stated correctly and implemented in the code(s) as intended. For example, I believe**

**While et al. (2017) has a typo in their thermal skin layer description.  They also use the Artale et al. (2002) scheme, but their notation (equation 1) introduces a constant beta, but the value they state (beta=864) does not match the value that would be derived from the Artale et al. (2002) (equations A1 and 3) or Tu & Tsuang (2005) (equations 1 and 6) descriptions!**

We double-checked the code several times, and I stress that we didn't touch anything in the cool skin scheme, that is already present within NEMO and is correctly coded.

**Line 220: No need for the line break**
Fixed.

**Line 250: Why does a1+a2+a3 not equal 1?  Why is the scheme designed like that? Is it designed differently to equation (13), where the a_i's sum to 1 (as expected)?**

We beg your pardon, but the comment is not fully clear to us. Following Zeng and Beljaars 2005, and therefore Soloviev 1982, we have the following values:
a1=0.28,
a2=0.27,
a3=0.45.

**Line 259-261 (also Table 1 caption): Can you be more careful here.  Paulson & Simpson (1981) propose/use the 9-band parameterization and take the coefficients from Defant (1961) based on clear water data.  Soloviev and Schlussel (1996) provide a variety of values for b_1 (your notation) based on different water types (from Jerlov, 1976).  However, without knowing what the Jerlov water type is this is no use. Instead, what has often been done is to take the mean value of b_1 for Jerlov types I, IA, IB, II, and III.  Hence, b_1 becomes 0.1488.**

**Note, that Gentemann et al. (2009) actually keeps the original Paulson & Simpson value for b_1 (i.e., pure-water), but includes solar angle in the parameterization (see their equation (14)) - is that a helpful modification and is it implemented today!? If not then the reference to Gentemann et al. (2009) is probably not helpful.**

Thanks for adding value with this comment, We fully agree with the reviewer and realized that the submitted version was not very clear in this regard. We inserted both of those lines, in the table caption as a reference to this.
Text modified from
"A formulation with 9 coefficients (reported in Table 2) has been proposed to include such effects (Soloviev and Schlussel, 1996, Gentemann et al., 2009) the first of them accounting for mean properties of I, IA, IB, II and III Jerlov's optical water types."
to
"Formulations with 9 coefficients (reported in Table 2) have been proposed to include such effects: for example  Soloviev and Schlussel, 1996 use a different coefficient for the first term depending on Jerlov's optical water type, while Gentemann et al., 2009 include solar angle in the parameterization, keeping the value of the of the first coefficient as in the case of pure-water. Without knowing what the Jerlov water type is, what is currently implemented in

NEMO is to take b_1 as the average between coefficients for I, IA, IB, II and III Jerlov's optical water types."
See LL 286-292 of the file with the track of the changes.
Also, we modified the caption of table 2 from
"Parameters for the Transmission coefficient following Soloviev and Schlüssel, 1996."
to
"Parameters for the Transmission coefficient following the formulation within NEMO (Soloviev and Schlüssel, 1996, Gentemann et al. 2009 and references therein), in which the first coefficients is the average between the one corresponding to I, IA, IB, II, and III Jerlov optical water types."

**Line 263: Is it right to call this ratio a "coefficient"?**

Thanks. We kept just "solar transmission" in the revised version.
See LL 293 of the file with the track of the changes.
**Line 266: "… depth at which transmission drops by 1/e from its surface value …" don't you mean "to 1/e" and not "by 1/e"!?, i.e., the depth it reaches when it decreases *to* ~37% of the surface value. (see also line 285 and line 303).**

Yes, I meant "to" and not "by". Corrected.
See LL 297, 323 and 482 of the file with the track of the changes.

**So the e-folding depth is 3m in the Soloviev (1982) scheme, but it is 1.6m in the Soloviev and Schlussel (1996) scheme – is that correct?  And it looks like using 3m in the S&S (1996) scheme takes you to 25% of the surface value (if I'm correctly reading and understanding your Figure 2a).**

Correct.

**The issue you are raising is not really whether d=3m is a good choice or not, but that the 9-band parameterization with fixed coefficients is not sufficiently accurate within the top 3 metres.  This is already well known.  Hence, we have solar transmission parameterizations whose coefficients/parameters change based on ocean color/chlorophyll data.  Key references include Morel & Antoine (1994), Ohlmann & Siegel (2000), and Lee et al. (2005).  These parameterizations have all previously been used and assessed in diurnal warming modelling studies.**

We see the concern, however, to the best of our knowledge, and as pointed out in the provided comments at the beginning of this report,  the idea of finding the e-folding depth, instead of cutting the solar transmission to a constant reference depth is a new approach that this study discuss it.
See LL 314-315, where we modified the text from
"This would give a constant transmission throughout the basin, but with a spatially and temporally varying e-folding depth and defines our new prognostic scheme for skin SST warm layer calculation."
to

"This would give a constant transmission throughout the basin, but with a spatially and temporally varying e-folding depth and defines our new prognostic scheme for skin SST warm layer calculation, thus embedding in it the ocean color information coming from Chl-a."

**Line 272-276: Please explicitly state this parameterization with formula and table of values as necessary. Determining this e-folding depth seems to be your key update to the prognostic model, but you say very little about it. Have other studies verified/assessed its accuracy? Can we have the full details, use the supplemental materials if necessary.**

Added the modified fortran source files of NEMO in the Zenodo repo.

**If I understand correctly, you are using this approach to estimate the e-folding depth and then using this value of d in equation (10), basically no longer requiring the 9-band parameterization (i.e., do not use the right-hand side of equation (13)).**

Yes it is true. We estimate the value of d from eqn. 13 (see comments above), using it into equation 10.

**You have shown that equation (13) is not very good at predicting the e-folding depth, i.e., the fixed transmission coefficients typically used are not very good at predicting solar absorption within the diurnal layer. For the Mediterranean Sea at least, this leads in an underestimation of solar penetration (as you note on line 271 and we see in Figure 2), making the simulated diurnal layer too warm on average. (And I'm trying to link this understanding to your results/figures – so if the simulated diurnal layer is too thin and too warm it would limit the development of large diurnal cycles in SST, hence why the magnitude of the diurnal cycle is increased with the new scheme – am I on the right track? It would be helpful to spell out some interpretation/explanation in the results/summary sections to help the reader understand what is going on).**

We added in the conclusion some words which help clarifying this point and providing a conclusive remark on the study. "Our interpretation is that within modradnemo, the chl-interactive e-folding depths allow, where and when necessary, the warm layer to become a little deeper than in the already existing scheme (nemoskwrite), depending on chl-variations. For these space-time points, solar penetration is increased and so it tends to make warmer the warm layer."
See LL 499-502 of the file with the track of the changes.

**How does this approach compare to the schemes that use non-invariant transmission coefficients; e.g., the Ohlmann & Siegel (2000) and Lee et al. (2005) parameterizations? Why not just use a state-of-the-art parameterization to begin with?**

Ohlmann and Siegel 2000 papers use the HYDROLIGHT radiative transfer model applied to a variety of factors, including chl, solar zenith angle and so on, as also done by Lee et al. 2005 with their model. Without any doubt, integrating their models into NEMO would improve a lot the situation, but we preferred in this paper to start simple and introduce only slight modifications to the already existing scheme, which can be regarded as our state-of-the-art

parametrization to begin with. Our work it is a step in their direction indeed, and these inclusions are underway for future developments of our work. We added a sentence on that in the concluding section

"Furthermore, testing the implementation within NEMO of more sophisticated radiative transfer models (such as the one of Ohlmann and Siegel 2000), or the development of deep learning based parameterizations are underway as future research efforts."
See LL 505-507.

**Section 3.3.1: Can you specify in the text the temporal frequency in which you determine the e-folding depths, and how this average is obtained (is it the mean of daily values over the period 2019-2021!?). Is the intention to use these seasonal e-folding depths for all future uses of the new scheme; i.e., you only need to compute this dataset once, a prior? Although obviously would need to be done again for a different region of the ocean.**

We left unspecified this aspect because it depends on how you set up NEMO. In our case, NEMO uses a chl-climatology of daily values. So the e-folding depth is calculated on that basis. But in principle, it can be adapted to other setups as well. We specified at the end of the section our protocol.
See LL 325-326 of the file with the track of the changes.

**Line 280: This is the first mention of Takaya et al., (2010) and assumes readers are already aware of what those refinements are. Please briefly explain here or earlier the changes/improvements T10 made to ZB05.**

We Modified the sentence from "Everything else is left unchanged, both the refinements of Takaya et al., 2010 (T10 hereafter) and the A02 model for cool skin." to
"Everything else is left unchanged, both the refinements of Takaya et al., 2010, which include the effect of Langmuir circulation and a modification of the Monin-Obukhov similarity function under stable conditions (T10 hereafter), and the A02 model for cool skin, which has been demonstrated to improve the scheme respectively under wavy and windy conditions."
See LL 315-318 of the file with the track of the changes.

**Also, just to confirm WRF uses ZB05 without T10 and A10 (your case 2: wrfskin), but otherwise is identical to the NEMO implementation of ZB05 (your case 3: nemoskwrite). Correct?**

Correct, wrfskin uses ZB05 without the modifications in A02 and T10, as explained in Table 1.

**Section 3.4: Table 1 is helpful and needed as the text description is not always so clear at first reading.**

Thanks, indeed we conceived it to improve readiness of the section.

**Line 299 (also Table 1): As mentioned earlier, I think the Gentemann et al. (2009) reference is potentially misleading, as I don't believe the scheme is what G09 state/use in their paper.**

This reference is present also in the NEMO original code, so we just reported it. Nothing changed.

**Line 305: "… (see section 3) below."  Do you mean "(see section 3.3.1)" which is actually above!**

Thanks. We cancelled the word "below", it was a typo from a previous version of the manuscript with another order for sections.
See LL 344 of the file with the track of the changes.

**Line 314-315: suggest rewording "… simulations outputs with data … assess methods performances …"**

Reworded from
"In this section, we compare simulations outputs with data from different sources (see section 2), to assess methods performances and impacts of our modifications."
to
"In this section, we present skill scores against satellite, drifters and temperature profiles data (see section 2) from the set of the simulations performed, aimed at characterizing the impacts of our modified skin SST scheme."
See LL 353-354 of the file with the track of the changes.

**Equation (14): Do you mean to start the index at i=0? If so, wouldn't the divisor be N_seas + 1?  Also, I think it's better/correct to write max SST – min SST (i.e., don't pull SST(h_i) outside the bracket as you currently have).**

Thanks, we have corrected eqn 14 according to your suggestions.
See LL 366 of the file with the track of the changes.

**Line 331: "… for every of the ..." --> "… for all of the …"**
.
Thanks, done.
See LL 373 of the file with the track of the changes

**Line 342: "over estimation of the wrfskin … except winter" Isn't the modrdnemo even more over estimated than wrfskin!?**

Yes, we modified the text from
"The spatial average over the whole Mediterranean domain is shown in figure 4, confirming the general underestimation of the control run and the overestimation of the wrfskin (ZB05 scheme) in all seasons except winter."
to
"The spatial average over the whole Mediterranean domain is shown in figure 4, confirming the general underestimation of the control run and the overestimation of the wrfskin and modradnemo in all seasons except winter."
See LL 383-385 of the file with the track of the changes

**Line 344: Your wording should be more careful in the first line of this paragraph.**

Wording reviewed, from
"Spatially averaging highlights that our modification brings improvement, especially during wintertime, …"
to
"Computing spatial averages highlights that modradnemo slightly improves the mean diurnal warming amplitude signal during wintertime, …"
See LL 386-387 of the file with the track of the changes

**Line 350-352: modradnemo looks similar, but slightly worst, to me compared to nemoskwrite (except for May)!?  Maybe a table of values would also complement the figure.**

We added the numbers on the plot instead of the colorbar.
Rephrased lines from
"over the old one (nemoskwrite) especially in May, but not in June, despite in the rest of the period the amplitude of the bias is slightly reduced."
to
"...over the old one (nemoskwrite) especially in May, but not in April, June, July, August and September, despite in the rest of the period the amplitude of the bias is slightly reduced."
See LL 393-395 of the file with the track of the changes

**Section 4.4: Can you try and explain why you see these changes, why are the different methods influencing the results?  I'm trying to make sense of these results (figures 4 and 5) based on what I see in Figure 2. Please can you help the reader in this regard.**

Given the changes the different methods determine on the upper ocean temperatures, impacts on air-sea heat fluxes and related phenomena (e.g. MLD variability) are expected. In this section, we wanted to test whether the impact is small or large, and we found that in general there's no great impact if one chooses one method or the other. Therefore, we did not deepen the discussion regarding this aspect, particularly, that we simulate a short period (2019-2021), which limits characterizing climate-relevant phenomena (MLD variability). We added a comment on this, adding a sentence
"It may seem from this picture that there's not such a huge change to prefer one method over the other considered in this paper, and this may also be because of the short period simulated (2019-2021)"
See LL 439-441.

**Line 359: "can be also" --> "can also be"**

The correction is considered in the revised version.
See LL 402 of the file with the track of the changes

**Supplemental Materials:**

**The first term 0.065 modifies Fairall et al. (1996)'s value of 0.137 (The Fairall value of 0.137 is obtained by summing a_5 to a_9 from your Table 2).  Wick et al. (2005)**

**suggested using 0.067 (0.137-0.07) as an ad-hoc adjustment to provide results that were comparable to the better Ohlmann & Siegel (2000) parameterization. However, I am unsure why the value of 0.065 is stated in Zeng & Beljaars (2005) (also in Zhang el al. (2021) so perhaps has become established) and repeated here in your supplemental materials. Which value is used in the actual code?**

See the link to the source files in the updated version of the Zenodo repo.

**This number can make a difference. For example, suppose delta was 1mm, then $f_s$ (equation (17)) in Fairall et al. (1996) would give 0.1009; i.e., 10% of solar radiation absorbed (or using the reduced value in Wick et al. (2005) it would be 0.0309, or using the value from Zeng & Beljaars (2005) 0.0289). I suppose I don't really get why this approximation is even needed in the first place. Why not just plug delta into the full Paulson & Simpson (1981) 9-band parameterization? You have the complete expression on line 202. (This is in fact is what Paulson&Simpson (1981) did see their equation (6)). Wouldn't a chlorophyll-based scheme help here as well?**

**The delta equation in your supplemental materials, although identical to equation (6) in Zeng & Beljaars (2005), is not the same as the referenced Fairall el al. (1996) equation (14) [note that this is an equation for lambda not delta!], I believe there must be a typo in Zeng & Beljaars (2005). The correct Fairall et al (1996) equation is reproduced in Tu and Tsuang (2005) (see their equation (4)) and Zheng et al. (2021) (see their equation 4). Please clarify. And perhaps more importantly is the correct expression used in the actual code(s)!**

We referred to Zeng and Beljaars 2005 derivation. See Changes in the Supplementary Material

---

## Author Comment (AC3)

**Response to Reviewer 2 comments**

We wish to thank Reviewer 2 for dedicating time to carefully read our work and providing his/her feedback. We sincerely think his detailed comments helped us to improve the manuscript. Here it follows a point-by-point response to the reviewer's report (text in bold denotes the provided comments, while normal text denotes our response), associating with the revised manuscript with the track of the changes.

**In this paper Authors propose an updated scheme to simulate heat propagation at the ocean-atmosphere interface. The importance of the task is related to heat budget estimates, assimilation of satellite data.**

**The study proposes an interesting and important upgrade, but in my opinion there are points that could be improved:**

- **Description of the model/new parameterization. It would be useful to add a diagram illustrating the layers, the cold skin layer and the warm layer. In the description of the model (e.g. Eq 5 and 10) it would be useful to understand what the prognostic variables are and how this parameterization is related to the OGCM internal variables. (e.g. Nemo potential temperature). Nemo vertical discretization in the upper layer could be also shown in the diagram.**

We can provide a schematic diagram of the cool skin and the warm layer (see respectively panels a and b of the figure below), even if it is just a re-adaptation of the sketch given in Donlon et al., 2007.
Since the cool skin temperature difference is time-independent, that is the diagnostic part of the scheme, while the warm layer temperature difference is the prognostic variable, since its determination requires temperature difference both at time t and at the preceding time step.
By default, the parameterization is conceived to simply diagnose the skin SST within a simulation, without actually entering in the dynamics. For each of the simulations, we substituted SST in the first model level with the skin SST calculated from each scheme in the coupled ocean-atmosphere model. In the diagram below you can also notice that vertical levels within NEMO are not equally spaced (we didn't report them perfectly in scale with the underlying y-axis - the figure is just qualitative).
Not in order, we made the following modifications to the manuscript:
- Added the information about the unevenly spaced vertical levels of NEMO in the model description part, changing the text from
"...72 vertical levels and a timestep…"
to
"...72 unevenly spaced vertical levels (the first and the last being respectively about 0.5m and 200m thick) and a timestep…"
See LL 188-190.
- Added comments on diagnostic/prognostic variables within the schemes. We modified text from
"...gets absorbed within the cool skin."
to
"...gets absorbed within the cool skin. Being time-independent, the cool skin temperature difference is a diagnostic variable in the scheme."
See LL 231-232.
And from
"...for the cool skin and warm layer respectively."

to

"...for the cool skin and warm layer respectively. Being time dependent, the determination of the warm layer temperature difference at time t requires the knowledge of the one at the previous time step, and thus is the prognostic variable in the scheme."

See LL 278-280.

- Added comments on the protocol used (substitution of the SST with the skin SST in the coupled simulations). Text changed from

"...(provided in Table 1)."

to

"...(provided in Table 1). In cases where a skin SST scheme is active, we substitute the SST, i.e. temperature on the first NEMO level, with the skin SST coming out from the scheme."

See LL 338-339.

- We added the sketch in the introduction section (see Figure 1), with the caption

[revised manuscript text omitted]

See LL 209-211.

**P6 Line 191 "assuming this constant" which constant, maybe the const in eq 2?**

**Please clarify.**

Yes, exactly.

**P7 Lines 212,214 please put units of measure of the symbols.**

Added, text modified to

"where $\lambda$ is the Saunders' proportionality constant, $Q$ ($W\ m^{-2}$) has already been defined above, $\tau/\rho_w$ ($m^2\ s^{-2}$) is the kinematic stress (ratio between wind stress module and seawater density), and $v_w$ ($m^2 s^{-1}$) $k_w$ ($m^2 s^{-1}$) are respectively the kinematic viscosity and thermal conductivity of seawater."

We left unspecified the dimensions of the Saunders' constant because they are given by dimensional analysis.

See LL 239-243.

**P7 Line 230 Eq 8 is not clear. Please write better the argument of \phi function.**

modified text from

"...depending on the sign of its argument:..."

to

"...depending on the sign of its argument, which is the ratio of the vertical coordinate to the Monin Obukhov length $L$:..."

See LL 264-265.

**P8 Line 241 "Assuming a temperature of dependence"**

**Please explain better.**

Sorry, there was a typo. Eliminated the word "of" give sense to the sentence. Changed from

"...Assuming a temperature of dependence.."

to

"...Assuming a temperature dependence…"

See LL 273.

**P8 Lines 245,246 I would specify if the equations are coupled e.g "In ZB05 scheme (Zeng and Beljaars, 2005), eqs. (5, 10) are the coupled equations for the cool skin and warm layer respectively." As already reported above, it would be useful to make clear which prognostic variables are taken into account.**

Thanks, changed accordingly. Text modified from

"In ZB05 scheme (Zeng and Beljaars, 2005), eqs. (5, 10) are the equations for the cool skin and warm layer respectively"

to

"In ZB05 scheme (Zeng and Beljaars, 2005), eqs. (5, 10) are the coupled equations for the cool skin (diagnostic part) and warm layer (prognostic part) respectively."

See LL 277-278.

**P8 Line 246 " within this layer" which layer? Maybe "these layers"?**

Changed from

"within this layer"

to

"within the warm layer"

See LL 280-281.

**P12 Line 337 "Looking at the mean profile averaged over all grid points in the given area, the agreement is better for all simulations during summertime months, both for the eastern and the western region (see figs. 8c, 9c), showing in particular that the modradnemo simulation outperforms the nemoskwrite one"**

**I don't see where modradnemo simulation outperforms the nemoskwrite one, they seem equivalent in terms of skill, could authors be more quantitative? Some tabulated skill metric could be useful.**

We thank the reviewer for this observation. It can be noticed from vertical profiles that the purple curve (modradnemo) is slightly closer to the blue curve (EN4) than the red curve (nemoskwrite). This is especially true in the central part of the med sea for summertime months, also when you look at RMSE on the upper 15m (see figures 10d, 10e).

**P12 line 376 "On the other hand, in the western Mediterranean all simulations tend to overestimate the signal, with our modified scheme doing a better job. "**

**Could authors be more quantitative, adding some statistics to support their statements?**

Sentence extended. From

"...doing a better job."

to

"...doing a better job with respect to the nemoskwrite case, with an average profile which is about 0.4°C closer to the EN4 profile."

See LL 429-430.

**Figure 2 panel (a) the plot could be improved. I suggest putting the z axis vertically (with negative ticklabels for depth as used in equations).**

Thanks for the suggestion; however, we'd prefer to keep the plot as it is, mainly for three reasons. First, Solar transmission is a function of depth, calculated as a sum of exponentials (inverting this function would not be as straightforward as one can think at first impression). Second, like this, it highlights that depth is the independent variable, and therefore has been placed on the x-axis. Lastly, being a log-log plot negative values would make no sense, since the logarithm function is defined for positive values of its argument.

**Figure 10. Region 0 should be removed, it's not a sea region.**

Thanks, changed accordingly. See figure below:

[Figure]

---

## Author Response (AR2)

**Response to Reviewer #1 Technical comments:**

On behalf of all the authors of the manuscript, I wish to thank Reviewer #1 for dedicating further time and effort to the revision of our manuscript. His attention and care of details really added value to our work.

Here follows a point-by-point response to his comments (bold text denote the comments, plain one the answers – our line numbering refers to the clean version of the updated manuscript):

**The authors have engaged with the reviewer comments and made good steps to revise and improve their manuscript. I would like to see this published; however, I have a few remaining (minor) comments and corrections based on the revised manuscript.**

**The line numbering used below is from the track-changes version of the updated manuscript.**

**Line 19: "All existing methods …" I would be happier if this said "All existing prognostic methods …" as this is not true in general, as there are diurnal skin SST models/methods that do not truncated solar transmission, you are referring to prognostic warm-layer-only methods.**

Added the word prognostic, Line 19.

**Line 20: "… we develop a new scheme …" my understanding from what you have written is that it is not a \*new\* scheme but an option that is already coded within NEMO, your contribute is to implement it for the prognostic skin SST scheme. As such, I think it more appropriate to word as "… we implement a new scheme …".**

Accepted. Reworded from "develop" to "implement". See Line 21.

**Line 32-34: "More in detail, the new scheme reduces the RMSE on the top 15 m in the central Mediterranean for summertime months, compared to the one already implemented one within the ocean model."**

**To improve the English I suggest replacing with**

**"The new scheme reduces the RMSE in the top 15 m of the central Mediterranean for the summertime months compared to the scheme already implemented within the ocean model."**

Accepted. See Lines 31-32.

**Line 107: "… coefficients though (Gentemann et al 2009)." Should be "… coefficients through Gentemann et al 2009."**

Changed accordingly, see Line 105.

**Line 244: "… which however …" replace with "… although …"**

Changed accordingly, see Line 237.

**Line 251: No need for a new line for this last sentence.**

Changed accordingly, see Line 243.

**Line 297: "… of the of the …" replace with "… of the …"**

Thanks, changed accordingly. See Line 288.

**Line 333: "In out setup …" replace with "In our setup …"**

Thanks, changed accordingly. See Line 321.

**Figure 1: Your black line depicting the diurnal thermocline has the cool skin effect penetrating to a depth of around 1.5m, this should only be confined to your cool-skin (blue) layer which is the top 10\mu m. From the black line the figure also is indicating a cool-skin effect of over 2 degree C, which is much too large. Please revise the black line in this figure before publication.**

Modified. See the new figure 1 in the Figures section.

**Lines 584-587: Your panel numbers need to be updated from "2a" and "2b" to "3a" and "3b"**

Thanks, changed accordingly. See Lines 569-572.

**For panel 3a why does the green line not intercept the y-axis at exactly 1? (like the red line does).**

Since the curve is the sum of exponentials, both green and red lines intercept y-axis at 1 only at a depth of zero meters on the x-axis. This point cannot be shown on a log-log plot, so we have to cut at a certain (arbitrary) value, whose choice was dictated by the need to show e-folding depths for both curves. We added a sentence in the figure caption: "Note that the x-axis range does not start from 0 to allow a logarithmic representation of the depth", Lines 572-573.

**Line 609: Your panel numbers need to be updated from "6a, 6b, 6c, 6d" to "7a, 7b, 7c, 7d"**

Changed accordingly. See Line 594.

**Supplemental Materials: I don't have an updated version, but the equation "delta = …" should be "lambda = …". The equation you provide is actually an equation for the Saunders' constant (which is needed for your eqn (6)). To provide closure you will also need an expression that relates delta (coolskin depth) to lambda.**

Thanks, we didn't notice the typo in Zeng and Beljaars 2005. Fixed accordingly. See updated supplementary material.